# A Method and a Metric for GNN Reliance on Information from Features and Structure

## Abstract

Graph Neural Networks (GNNs) rely on both node-edge features and graph structure, but the relative use of these information sources is poorly understood. In many cases either features or structure contain more useful information, and in extreme cases one may inhibit learning, as in some tasks where models overfit on structural patterns. Understanding the balance of these information sources is therefore essential for strategic model design.

We introduce Noise-Noise Analysis to measure each source's contribution to model performance, along with the Noise-Noise Ratio Difference (NNRD) metric that quantifies whether a model is *feature-reliant* or *structure-reliant*. Through experiments on synthetic and real-world graph-classification datasets, we show that GCN, GAT, and GIN layers can all perform graph-less learning (ignoring structure when unhelpful), but only the GIN performs feature-less learning. All three architectures exhibit bias toward features over structure. Noise-Noise Analysis provides practitioners a fast tool to understand their models' information usage.

## 1 Introduction

Graphs are a highly flexible way to represent data in many fields of industry and science, including chemistry (Gilmer et al., 2017), infrastructure planning (Khodayar et al., 2019), biology (Li et al., 2022) and social network analysis (Davies et al., 2025). Many graph-learning tasks focus on predicting a graph-level property, extracting useful information from graphs for a given target, for example solubility in molecules (Wu et al., 2017). In this work, we focus exclusively on such graph-level prediction tasks, where the goal is to predict a single label for an entire graph. Information for these tasks is provided by both features attached to nodes and edges and the information encoded in the structure of the graph.

For learning tasks with data from multiple sources, one source frequently carries more information, or indeed more noise. Targeting the more useful information source, or a learner's capacity to correctly favour one source, is a crucial aspect of design and implementation. In this work we take the view that for graphs these information sources are features and structure.

Graph Neural Network (GNN) models parametrise message-passing (information passing between nodes), allowing flexible and expressive learning on graph data. The degree to which GNN models rely on either structural information through edges $E$ or through features $X$ during learning is highly varied. Some works target structural learning, aiming to improve performance by biasing towards the information encoded in patterns between nodes (Chen et al., 2021; Horn et al., 2021). Other works argue that the optimal solution may not include graph structure at all (Bechler-Speicher et al., 2024). In extreme cases a graph-learner should be able to entirely ignore an information source when it is not useful; the ability of a graph-learner to ignore information sources is its ignorance capacity (Bechler-Speicher et al., 2024). Whether a dataset and model combination leans towards structure or feature information is currently not clear.

**Contributions** Here we propose and demonstrate Noise-Noise Analysis, a method for characterising the balance of information between features and structure a model extracts in a given graph-learning task. We also introduce Noise-Noise Ratio Difference (NNRD), a quantitative metric based on Noise-Noise Analysis. Following a noising process along either features or structure, Noise-Noise Analysis and NNRD give an

intuitive value with which data publishers or users can tailor their approaches from the outset. By using Noise-Noise Analysis with pre-trained graph-learners, results are fast to calculate and intuitive to evaluate.

We design a set of synthetic datasets to test information use and ignorance capacity. Using Noise-Noise Analysis we show that parametrisation has a significant impact on a graph-learner's capacity to ignore information sources. Specifically, we show that:

**(i)** Regardless of parametrisation, message-passing layers are capable of graph-less learning.

**(ii)** Layers parametrised during message-passing do not perform feature-less learning.

**(iii)** All layers are capable of feature-less learning when structural information is included as a component of features through positional encodings.

We also apply Noise-Noise Analysis over a selection of real-world datasets, demonstrating that our findings from synthetic datasets carry over to real-world applications. We reproduce and extend the results from other works (Errica et al., 2022; Bechler-Speicher et al., 2024), showing that layers do not make use of structural information on some datasets unless it is included explicitly through positional encodings. We also provide an example of using Noise-Noise Analysis in model design over molecular benchmarks, showing how insight into the balance between feature and structure information can lead to improved performance.

**Organisation** In Section 2, we detail the technical background and related work necessary to understand this work. In Section 3, we introduce Noise-Noise Analysis and its companion metric NNRD. Next, in Section 4, we apply Noise-Noise Analysis over a wide range of synthetic and real-world datasets, then Sections 5 and 6 present and discuss these results. Finally, in Section 7, we conclude by summarising this work and its implications, as well as proposing future research directions.

## 2 Background and Related Work

**Graph Neural Networks** GNNs, or more specifically Message-Passing Neural Networks (MPNNs), parametrise message-passing and aggregation, allowing deep-learning techniques to be applied on graph data. Most GNNs parametrise either message-passing or aggregation, though some theoretical formulations assume that both are parametrised (Bechler-Speicher et al., 2024). For a dataset with nodes $V$, real features $X \in \mathbb{R}^{|V| \times D}$ and similar for edges, structure as an edgelist $E : \{(v_1, v_2), \ldots\}$, and graph labels $y$, graphs are $G : \{X, E\}$. When a given model aims to learn some mapping $f(G) \to \hat{y}$, in reality this is $f(X, E) \to \hat{y}$.

The usual graph learning assumption is that structure $E$ and features $X$ must be considered together for information to be useful; their combination has mutual information with a target $y$, $I(y; X, E)$. We assume that although for most tasks the combination of $X$ and $E$ information is most useful, both $X$ and $E$ contain useful information independently.

For a message-passing GNN, a node's state is updated as:

$$h_i^{k+1} = \text{Agg}^k \left[ \text{Mess}^k \left( h_i^k, h_j^k \right) | j \in N(i) \right], \tag{1}$$

where $h_v$ is the current state of node $v$, $k$ is the current message-passing layer, $N(v)$ is the neighbourhood of node $v$, Mess is a message-passing function, and Agg is an aggregation function over those messages. Both Agg and Mess can be learnt functions for each layer, hence $\text{Mess}^k$ and $\text{Agg}^k$. We make use of three popular GNN layers; the Graph Convolutional Network (GCN) (Kipf & Welling, 2017), the Graph ATtention (GAT) layer (Veličković et al., 2018), and the Graph Isomorphism Network (GIN) (Xu et al., 2019). These three layers are widely used, and as each parametrises message-passing and aggregation differently, represent an archetypal set of MPNNs. Graph transformers (Zhang et al., 2025) and scalable GNNs use or emulate similar parameterisations (Rampášek et al., 2022), and the findings of this work are therefore relevant also to these architectures.

**Positional Encodings** Several works have demonstrated the limited capabilities of message-passing in expressing graph structures, with isomorphism a clear example (Xu et al., 2019). The implications of these

limitations are that local message-passing GNNs, like the GCN, GIN and GAT described above, may be unable to represent and therefore learn from the information in some structures. A popular remedy is to use positional encodings. Here some set of values are computed, aimed at representing the structure around a given node. These positional encodings can then be attached to a node alongside its features. This is an intuitive way of giving a learner explicit access to structural information through features. A simple example is random walks; from the target node, randomly traverse between neighbours, recording which nodes have been visited, with this set of visited nodes then a positional encoding.

## 2.1 Ignorance Capacity

Let $\mathcal{P}(f, \mathcal{D})$ denote the performance of model $f$ on dataset $\mathcal{D}$. For information source $K$, define:

**Definition 1.** *Model $f$ has perfect **ignorance capacity** for source $K$ if:*

$$\mathcal{P}(f, \mathcal{D}) = \mathcal{P}(f, \mathcal{D}^{\emptyset_K}) \tag{2}$$

*where $\mathcal{D}^{\emptyset_K}$ represents the dataset with information source $K$ replaced with uninformative content.*

**Definition 2.** *A model has **structure ignorance capacity** if it can maintain performance when graph structure $E$ is uninformative, $\mathcal{P}(f, (X_i, E_i, y_i)) \approx \mathcal{P}(f, (X_i, E_i^{\emptyset}, y_i))$, where $E_i^{\emptyset}$ represents random or uninformative graph structures.*

**Definition 3.** *A model has **feature ignorance capacity** if it can maintain performance when node/edge features $X$ are uninformative, $\mathcal{P}(f, (X_i, E_i, y_i)) \approx \mathcal{P}(f, (X_i^{\emptyset}, E_i, y_i))$, where $X_i^{\emptyset}$ represents random or uninformative features.*

Bechler-Speicher et al. (2024) investigate the ignorance capacity of MPNNs, formulating a GNN layer specifically as in Equation 3:

$$h_i^{k+1} = \sigma \left( W_1^k h_i^k + W_2^k \sum_{j \in N(i)} h_j^k + b^k \right). \tag{3}$$

$W_1^k$ and $W_2^k$ here are learnable weight matrixes, termed the 'root' and 'topological' weights respectively. This treatment allows easy analysis of how a message-passing layer can ignore graph structure, with $W_2^k = 0$ resulting in 'graph-less' functions. Though flexible in theoretical treatments, for the archetypal layers we've described above, this formulation does not necessarily hold true.

The GCN and GAT layers have no explicit separation of the root weights and topological weights, so Bechler-Speicher et al. (2024) assume that they therefore cannot ignore graph structure. Each of these layers is widely used, and many other layers also don't follow the same formulation.

## 2.2 Information for graph-learners

Wu et al. (2020) explore the expressive power of GNNs, introducing the information bottleneck principle. Alon & Yahav (2021) perform similar analysis, with both works noting the difficulty of balancing different information sources during graph learning. More recent works have elaborated on the same theme, often emphasising that structural information can be difficult to effectively incorporate (Wu et al., 2023). Other works instead show that node information can be lost during GNN use, with original feature similarities distorted during message-passing and aggregation (Jin et al., 2021). Some works explicitly decouple features and structure, finding performance benefits despite the loss of information from their inter-relation (Wang et al., 2024; Davies et al., 2024), suggesting that for some tasks one source may actively inhibit learning from the other.

### 2.2.1 Theoretical Treatments

In settings without relying on Equation 3 it is much harder to prove theoretically that layers have the same ignorance capacity, or indeed verify within the general case of unspecified layers and datasets. In-particular, even with simple target labels, estimating information density across both features and structure for graphs is intractable. Instead in this work we explore empirically the ignorance capacity of these commonly used message-passing layers. We evaluate the method and metric described below empirically before presenting results, and demonstrate that they are consistent w.r.t. key graph properties within our datasets, providing greater trust in our findings and conclusions. This experimentation-first approach yields useful findings, with these findings then strong areas for in-depth theoretical analysis, and can be applied to any given graph-learner and dataset.

## 3  Noise-Noise Analysis

The balance between structural and feature information, and how to compromise between them, is an open issue. As the former concern must be addressed before the latter, we propose an analysis method to interrogate how a graph-learner weights between feature and structure. We also include a quantitative metric built on the same analysis method.

In the absence of features, structural information is still present, i.e. $I(y; E) \geq 0$, in the same vein features contain their own useful information, $I(y; X) \geq 0$, and we assume the two are not evenly balanced, $I(y; X) \neq I(y; E)$. Here we propose **Noise-Noise Analysis** as a method for investigating this imbalance with respect to model information use. By degrading the useful information in either $X$ and $E$, the degree to which performance relies on one or the other should be apparent. A caveat is that degrading either channel also degrades their joint mutual information $I(y; X, E)$. NNRD therefore reflects a model's relative reliance on each channel, rather than their fully isolated contributions. The Coupled dataset probes this case directly, requiring integration of both sources for accurate prediction.

Consider some destructive noising process $N_t(\cdot)$, that degrades the useful information in either $X$ or $E$ across a dataset. $N_0(x) = x$, with no useful information after $N_T(\cdot)$. Some imperfect model produces predictions $f(X, E) \to \hat{y}$. Performance is some $\mathcal{P}(y, \hat{y})$ between real and predicted values. After sampling $\mathcal{P}(y, f(N_t(X), N_t(E)) \to \hat{y}$ the balance of useful information in $X$ and $E$ should be observable.

Here we denote $\mathcal{P}_X(t)$ the performance of a given model on a task at some noising step $t$ for noise over features. Similarly $\mathcal{P}_E(t)$ is performance at structural noise step $t$. $\mathcal{P}_X(t)$ and $\mathcal{P}_E(t)$ should both decrease together with $t$, as information for downstream tasks is necessarily removed. If useful information is equally balanced between structure and features, they will descend at the same rate. If not, one will fall faster than the other, with increasing performance in the case of overfitting on one information source.

We can normalise performance metrics $\mathcal{P}$ between 0 and 1, at least within the test-set. Using these curves between noise and performance we define a metric, Noise-Noise Difference Ratio (NNRD), which measures the relative reliance of a graph-learner on structure and feature information.

**Definition 4.** *In the continuous $t$ case, with $0 \leq t \leq T = 1$, using a log ratio to preserve symmetry, and treating $t$ as a uniformly sampled random variable:*

$$\text{NNRD} = \mathbf{E}_{t \sim U(0,1)} \log \left[ \frac{\mathcal{P}_X(t)}{\mathcal{P}_E(t)} \right] \simeq \int_0^{T=1} \log \left( \frac{\mathcal{P}_X(t)}{\mathcal{P}_E(t)} \right) dt. \tag{4}$$

**If** $\log \mathcal{P}_X(t) > \log \mathcal{P}_E(t)$, NNRD is positive, indicating that structure noise has a larger impact than feature.

**If** $\log \mathcal{P}_X(t) < \log \mathcal{P}_E(t)$, NNRD is negative, indicating that feature noise has a larger impact than structure.

**If** $\log \mathcal{P}_X(t) = \log \mathcal{P}_E(t)$, NNRD is zero, indicating that performance relies equally on features and structure.

Here we expect higher $\mathcal{P}$ indicates better performance. In practice it's necessary to evaluate at discrete timesteps $T = \{t_0, t_1, \ldots\}$, $t_i \in [0, 1]$:

$$\text{NNRD} = \frac{1}{|T|} \sum_{t \in T} \log \left( \frac{\mathcal{P}_X(t)}{\mathcal{P}_E(t)} \right). \tag{5}$$

This approximation makes the number of evaluated timesteps a hyper-parameter for NNRD, similar to approximations made in binning-based performance metrics (Silva Filho et al., 2023; Kängsepp et al., 2025). Equation 5 is the standard sample mean estimator, which converges to the true expectation as $|T| \to \infty$ (Glivenko-Cantelli theorem (Vaart, 1998)). For smooth curves, the error rate is $O(1/|T|^2)$. We provide a sensitivity analysis in the Appendix, showing minimal change in NNRD with varying timestep size, and no significant correlation between timestep size and deviation in NNRD values. In the Appendix we also provide guiding values for NNRD, with $|\text{NNRD}| = 0.307$ indicating complete ignorance over either features or structure, and $|\text{NNRD}| > 0.307$ indicating overfitting on either features or structure.

We describe calculating NNRD in Algorithm 1. A graph learner $f$ is applied on a dataset $D$, which contains node/edge features $X$, edges $E$, and labels $y$ for each $g \in D$, with performance measured by a metric $\mathcal{P}$. For each noise level $t_i \in T$, we apply our noising functions FeatureNoise and StructureNoise in lines 6 and 7, and measure performance $\mathcal{P}_X(t)$ and $\mathcal{P}_E(t)$ in lines 8 and 9. NNRD is then calculated using their log-ratio at each noise level in line 11.

---

**Algorithm 1** Noise-Noise Analysis

1: **Input:** Dataset $D = \{(X_i, E_i, y_i) \mid i = 1, \ldots, |D|\}$
2: **Input:** Pre-trained graph learner $f$
3: **Input:** Noise levels $T = \{t_0, t_1, \ldots, t_n\}$ where $t_0 = 0$, $t_n = 1$
4: **Input:** Performance metric $\mathcal{P}$ (higher is better)
5: **for** each $t$ in $T$ **do**
6: $\quad D_X^t \leftarrow \text{FeatureNoise}(D, t)$ {Apply feature noise}
7: $\quad D_E^t \leftarrow \text{StructureNoise}(D, t)$ {Apply structure noise}
8: $\quad \mathcal{P}_X(t) \leftarrow \mathcal{P}(f(D_X^t))$
9: $\quad \mathcal{P}_E(t) \leftarrow \mathcal{P}(f(D_E^t))$
10: **end for**
11: $\text{NNRD} \leftarrow \frac{1}{|T|} \sum_{t \in T} \log \left( \frac{\mathcal{P}_X(t)}{\mathcal{P}_E(t)} \right)$
12: **Return** NNRD

---

By applying Noise-Noise Analysis over a hold-out set, the degree to which these convolutions rely on feature or structure information is evident. This means that noise-noise-analysis can be applied to arbitrary pre-trained models, without the need to re-train models.

## 3.1 Noise Functions

In order to directly compare performance at increasing noise levels, the proportional information from features and structure should degrade at an equal rate:

$$\frac{\partial \left[ I(\text{feat.})/I_{\max}(\text{feat.}) \right]}{\partial t} \simeq \frac{\partial \left[ I(\text{struc.})/I_{\max}(\text{struc.}) \right]}{\partial t}. \tag{6}$$

Within-sample noise functions, for example feature perturbation and edge-swapping, do not necessarily destroy information at equal or consistent rates. As an example, a single hydroxyl group drastically changes a molecule's properties, and a single broken edge might remove a ring structure.

Even if non-consistency is acceptable, there is no guarantee that noising functions of this type would destroy information at equal rates. Indeed it would be very hard to prove that they do, given the complexities of comparing graph structures to highly diverse node and edge features. As such Equation 6 would also not hold with these commonly used noising functions. The formulation here is for noising functions over graph

*datasets*, not over individual graphs. This lets us use the following noising functions, that remove information from entire graphs, essentially an all-or-nothing approach. With our structure noise, these two noise functions satisfy Equation 6, and performance curves with increasing noise on features or structure can be directly compared. We theoretically justify these noise functions in the Appendix. The theoretical justification for why graph-level replacement achieves equal proportional information loss, while within-sample methods do not, is provided in Appendix B (Theorem 3).

**Structure Replacement** Consider a graph dataset $\mathcal{D}$, and a noise on structures $N_t(E_i \in \mathcal{D})$. Information is complete at $N_0(E_i) = E_i$, with no structural information remaining at $t = T$, $I(y; N_T(E_i)) = 0$. Here our noising function is to replace a proportion of graphs $g = \{g_1, g_2, \ldots\} \in \mathcal{D}$ with Erdos-Renyi graphs of the same number of edges as the original graph. The number of graphs replaced is $|g| = t \cdot |\mathcal{D}|$, with the graphs to be replaced randomly sampled from $D$. This lets us maintain the original edge features in the graph, as the number of edges is constant. As we show in the Appendix, this noise function satisfies our constraints where the number of nodes $|V|$ and edges $|E|$, and quantities based on this counting, are not strongly informative.

**Feature Replacement** In a similar vein to our structure noise, consider a noising function on features $N_t(X_i \in \mathcal{D})$ with $N_0(X_i) = X_i$ and $t = T$, $I(y; N_T(X_i)) = 0$. We again take a proportion of graphs $g = \{g_1, g_2, \ldots\} \in \mathcal{D}$, $|d| = t \cdot |\mathcal{D}|$ and replace their features with noise sampled uniformly within the per-feature range of the training set. In the same manner that we preserve density with our structure noise, here we preserve range.

In datasets where original structure is close to ER graphs, or original features are close in distribution to the noise generator, the noise functions still replace the original information entirely — any correlation with labels is lost upon replacement. NNRD will correctly reflect low reliance on the noised channel in such cases, which is an accurate characterisation of the dataset rather than a methodological failure. We note that random node features have been shown to increase GNN expressiveness (Sato et al., 2021). However, as noise is applied only at test time to a pre-trained model, any such expressiveness gains cannot be exploited — the model's weights are fixed and were not trained to make use of random features.

## 4 Experiments

To demonstrate Noise-Noise Analysis we employ real-world and synthetic datasets in our experiments. As discussed above, our noise functions are restricted to a graph-level, which in turn leads us to focus on graph-level tasks instead of node or edge-level. Crucially, the unsuitability of within-sample noise also precludes our approach from application to most node-level or edge-level tasks, though given the same mechanism for a given MPNN layer in both settings, our findings should apply here too. Our synthetic datasets are designed to probe how effectively graph-learners can extract information in extremes of ignorance and understanding. The real world datasets serve two purposes; first we reproduce the findings of prior works using NNRD, then we give an example of using NNRD in model design through molecular benchmarks.

We use ten different noise levels from $t = 0$ to $t = 1$, and the above noise functions, with each training run of 50 epochs conducted 5 times to generate error bounds. This repeated training would not be necessary in-practise with larger pre-trained graph-learners. As models here we use 3-layer GNNs of 100 hidden units per layer, with GCN, GIN or GAT layers. The learning rate is set to 0.001 with a batch size of 256 graphs. These hyperparameters are chosen to match Bechler-Speicher et al. (2024) to allow direct comparison. Appendix E demonstrates that NNRD is stable across variations in feature dimensionality, graph size and density, providing evidence of robustness to non-task-relevant hyperparameter choices. Where positional encodings are included we use random walks with restarts, with dimensionality 16 and the chance of restart 0.2.

### 4.1 Synthetic Datasets

In order to explore the ignorance capacity of common GNNs, and to verify the efficacy of NNRD, we design synthetic datasets where the source of useful information is directly controllable. In our synthetic datasets, graph structures are sampled from two distributions, ring ladders $S = L$ or Erdos-Renyi (ER) graphs $S = R$ of corresponding density. Each graph has node and edge features of either the integer $x = 1$ or $x = -1$

for each node and edge. By crafting different sets of labels for the graphs in these datasets, we are able to directly control the source of useful information. For each dataset we sample an equal number of ring and ER graphs, and an equal number of graphs with $x = 1$ and $x = -1$. For each dataset we produce 3200 graphs, with a (0.7, 0.2, 0.1) split. We build our synthetic datasets as follows:

**Easy (Either Source)** Here both features and structure point to the same label, meaning that either or both can provide the information necessary for the task. This means that, with an ideal model, we expect no performance drop at complete noise on either features or structure. We include no examples where $S = L, x = 1$ or $S = R, x = -1$. **Feature (Single-Source)** Here only features point to the graph label, meaning that only features provide useful information, $I(y; E) = 0$. As such we expect that structure noise has no influence on performance. **Structure (Single-Source)** Graphs are ring ladders or ER graphs as above. The label is determined solely by the structural class (ring ladder $\rightarrow$ 1, ER $\rightarrow$ 0). Node and edge features are assigned $+1$ or $-1$ uniformly at random, independently of the label, ensuring $I(y; X) = 0$. Only structural information is predictive of the label. As such we expect that feature noise has no influence on performance. **Coupled (Both Sources)** Graphs are ring ladders or ER graphs as above, with features of $+1$ or $-1$. The label is positive when the structural class and the feature value agree (i.e. ring ladder with $+1$, or ER with $-1$), and negative otherwise. Neither source alone carries information about the label, but their combination is fully predictive, requiring the model to integrate both channels.

A visual illustration of the labelling schemes across all graph-level datasets is provided in Figure 10.

| Layer | Dataset | Base | Structure | Feature | F/S | NNRD |
|---|---|---|---|---|---|---|
| GCN | Easy | 1.0±0.0 | 1.0±0.0 | 0.5±0.0 | 0.5 | −0.22 |
| | Feature | 1.0±0.0 | 1.0±0.0 | 0.51±0.03 | 0.51 | −0.22 |
| | Structure | 1.0±0.0 | 0.5±0.0 | 1.0±0.0 | 2.0 | 0.16 |
| | Coupled | 1.0±0.0 | 0.49±0.01 | 0.51±0.03 | 1.0 | 0.092 |
| GIN | Easy | 1.0±0.0 | 1.0±0.0 | 0.59±0.01 | 0.59 | −0.22 |
| | Feature | 1.0±0.0 | 1.0±0.0 | 0.51±0.03 | 0.51 | −0.22 |
| | Structure | 1.0±0.0 | 0.5±0.0 | 1.0±0.0 | 2.0 | 0.31 |
| | Coupled | 1.0±0.0 | 0.5±0.0 | 0.49±0.01 | 0.98 | 0.16 |
| GAT | Easy | 1.0±0.0 | 1.0±0.0 | 0.61±0.01 | 0.61 | −0.22 |
| | Feature | 1.0±0.0 | 1.0±0.0 | 0.49±0.03 | 0.49 | −0.21 |
| | Structure | 0.51±0.0 | 0.51±0.00 | 0.3±0.0 | 0.59 | −0.19 |
| | Coupled | 0.58±0.0 | 0.52±0.00 | 0.5±0.0 | 0.96 | −0.16 |

Table 1: Performance for GCN, GIN and GAT models on our synthetic datasets. We report the performance with no noise (Base), at maximum structure noise (Struc.), at maximum feature noise (Feat.), the ratio of these performances (F/S) and the respective NNRD score.

## 4.2 Real-World Datasets

We apply Noise-Noise Analysis to several real-world datasets. We begin with the Enzymes and Proteins datasets, both of which have been used as examples of un-informative structure (Errica et al., 2022; Bechler-Speicher et al., 2024). Next, we demonstrate how Noise-Noise Analysis can be used in model design, taking molecular benchmarks as an example.

**Enzymes and Proteins** Bechler-Speicher et al. (2024) cite a result from Errica et al. (2022) regarding the Enzymes and Proteins TU datasets. Specifically, they show that their GNNs (which follow Equation 3) overfit to structure, and demonstrate that by replacing graph structures with regular or random graphs, performance benefits are possible. Here we re-produce that result, within the NNRD setting, i.e. in evaluating what convolutions are learnt from the original dataset. As we do not expect perfect performance, we train and evaluate models five times to establish the range at base.

**Molecules** Lastly we apply Noise-Noise Analysis over the MoleculeNet datasets (Wu et al., 2018), focussing on binary classification for clarity in interpretation. We focus on the hiv and tox21 datasets, which are

the largest of these single-target classification datasets. For the hiv dataset the task is predicting whether a molecule inhibits HIV replication, and for tox21, the task is predicting whether a molecule is toxic in humans. We use these datasets to show how Noise-Noise Analysis can be employed to make informed choices, simulating a researcher iteratively improving their model design. We assume that the user begins with a GCN model, the most simply parameterised of those we employ in this paper.

## 5 Results

Here we present our experimental findings, with further results and visualisations in the Appendix. As discussed above, theoretical treatments of the extremely general case presented here, where neither model nor dataset can be specified, are not feasible. This is particularly true when real layers, instead of the simplified model in Equation 3, are considered.

Instead, we validate the required properties of a strong metric (consistency across datasets, invariance with non-relevant variation). Specifically, we make use of our own synthetic datasets as cases where information sources are well-understood, vary properties not relevant to this information, then verify statistically that NNRD is stable. The results of this analysis can be found in Appendix Section E, and demonstrate the safety of employing our method and metric. On no dataset does a sample GCN, with errors calculated over five retraining runs, produce a significant variation as feature dimensionality, graph node count or graph density vary. The specific varied properties are node count, feature dimensionality and density where appropriate, with Table 2 showing the results of this analysis.

| Information Source | Nuisance Parameter | CV | K-W p-value | Levene p-value |
|---|---|---|---|---|
| Structure | Feature Dim. | 0.022 | 0.815 | 0.959 |
| | Nodes $|V|$ | 0.007 | 0.995 | 0.514 |
| Feature | Feature Dim. | 0.016 | 0.498 | 0.680 |
| | Nodes $|V|$ | 0.011 | 0.715 | 0.704 |
| | Density $\rho$ | 0.012 | 0.699 | 0.416 |

Table 2: The results of our consistency and robustness experiments for NNRD, using the Feature and Structure datasets. K-W tests that NNRD does not vary with nuisance parameters; over all tests it does not. Levene tests whether variance is equal across nuisance parameters; again NNRD's variance is consistent.

### 5.1 Synthetic Datasets

The results of our experiments with synthetic datasets can be found in Table 1.

**Structure Ignorance** On the Easy dataset, across all three message-passing layers, learnt convolutions are based only on features, demonstrated by the lack of performance change with structure noise, and negative NNRD values. Results for the Feature dataset, corroborating this finding, are functionally identical to those for the Easy dataset. Here we have empirically demonstrated that GNNs do in fact have the capacity to ignore structural information and learn graph-less convolutions. Further, we have shown that with the 'option' of becoming graph-less, feature-less or neither, all three layers become nearly graph-less.

**Feature Ignorance** We use the Structure dataset to explore the inverse; whether these graph layers have useful feature ignorance capacity. On the Structure dataset only the GIN performs true feature-less learning. The GAT layer reaches at-best random performance, failing to extract any structural information for the task, essentially becoming graph-less despite the lack of information in features. The GCN layer fares better, with perfect performance when the test-set is un-noised. It exhibits a non-linear trend with feature noise, recovering perfect performance when all feature-based information is removed. This indicates that its learnt convolutions are not strictly feature-less, but instead some more complex dynamic is at play. We perform the same analysis using accuracy, instead of ROC-AUC, for a GCN model on the Structure dataset. A visualisation is found in Figure 7 in the Appendix. Here we can see that performance is now as expected.

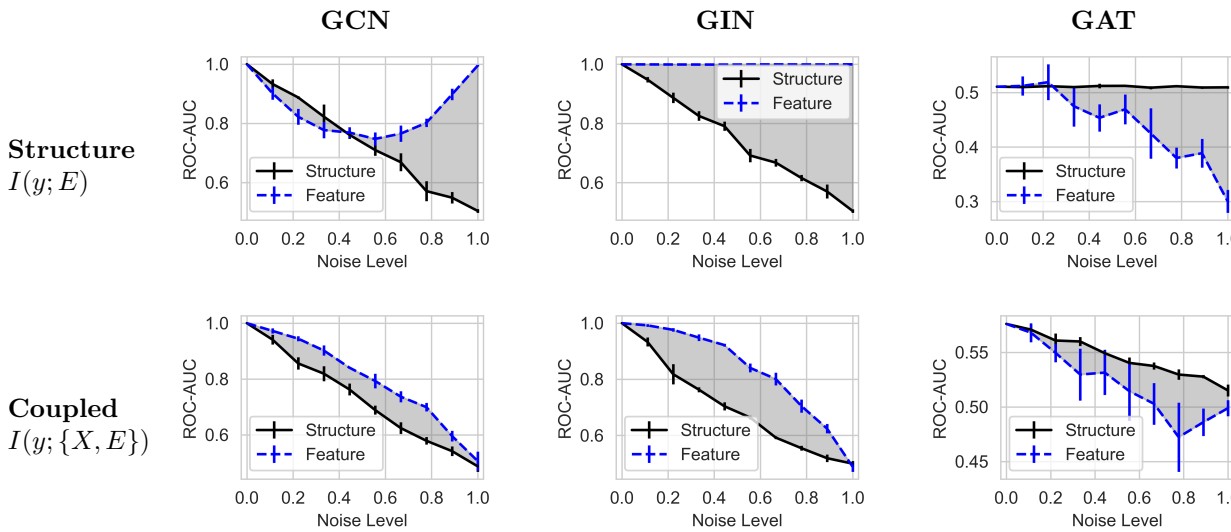

Figure 1: Performance variation for supervised training of our GCN, GIN and GAT models over synthetic datasets with increasing noise on structure and features in the test-set. All datasets report ROC-AUC.

**Combined Sources** The Coupled dataset tests a learner's ability to combine distinct information from features and structure. Both the GCN and GIN perform well, though NNRD for the GIN indicates a moderate bias towards structural information. Response from the GCN and GIN layers is larger from structural noise than feature noise, but the GAT shows the inverse, with large variation at each feature noise level. The GAT layer performs very poorly, with its base performance only slightly better than random, and has not effectively incorporated explicit information from graph structures.

### 5.1.1 Node and Edge-Level Tasks

The necessity of using whole-graph noise functions precludes application in most node and edge-level tasks. In the limited scope that all node labels are predicted at once, there is no need for induction requiring the labels on other nodes, allowing us to perform experiments on node classification. Results broadly mirror our graph-level findings, though on the Easy dataset all three models balance feature and structure use more evenly. The GAT demonstrates heavy overfitting on features on the Structure dataset. Full experimental detail and results are presented in Appendix G.

## 5.2 Enzymes & Proteins

Results for the Enzymes and Proteins datasets, both with and without positional encodings, can be found in Table 3, and visualisations in the Appendix. As expected, from both Bechler-Speicher et al. (2024)'s results and our earlier findings, all learnt convolutions rely more heavily on feature information than structural information when positional encodings are not included. Performance is effectively random here when all feature information is removed.

However, when positional encodings are included, all models perform better at base. They rely more heavily on information from the graph structure - demonstrated by increasing NNRD values - and have worse performance when all structural information is removed. This corroborates our earlier findings about these layers; they tend to rely on information from the feature channel more heavily than the structural channel, regardless of whether they are parameterised in message-passing or aggregation.

## 5.3 Molecules

Results on the hiv and tox21 datasets can be found in Table 4.

| Layer | Dataset | Base | Structure | Feature | F/S | NNRD |
|---|---|---|---|---|---|---|
| GCN | Enzymes | 0.69±0.00 | 0.67±0.03 | 0.48±0.01 | 0.72 | −0.33 |
| | Enz. (pos) | 0.72±0.02 | 0.63±0.03 | 0.49±0.05 | 0.78 | −0.23 |
| | Proteins | 0.72±0.03 | 0.74±0.05 | 0.49±0.05 | 0.66 | −0.37 |
| | Prot. (pos) | 0.78±0.04 | 0.75±0.03 | 0.6±0.0 | 0.8 | −0.28 |
| GIN | Enzymes | 0.78±0.03 | 0.72±0.02 | 0.48±0.03 | 0.67 | −0.33 |
| | Enz. (pos) | 0.82±0.05 | 0.65±0.03 | 0.53±0.01 | 0.82 | −0.25 |
| | Proteins | 0.76±0.04 | 0.81±0.04 | 0.4±0.0 | 0.49 | −0.31 |
| | Prot. (pos) | 0.82±0.02 | 0.71±0.04 | 0.51±0.12 | 0.72 | −0.26 |
| GAT | Enzymes | 0.71±0.02 | 0.68±0.03 | 0.52±0.05 | 0.76 | −0.38 |
| | Enz. (pos) | 0.7±0.0 | 0.58±0.03 | 0.49±0.02 | 0.84 | −0.16 |
| | Proteins | 0.7±0.0 | 0.69±0.03 | 0.51±0.04 | 0.74 | −0.31 |
| | Prot. (pos) | 0.8±0.0 | 0.75±0.03 | 0.67±0.07 | 0.89 | −0.16 |

Table 3: Performance for GCN, GIN and GAT models the Enzymes and Proteins TU datasets. We report the performance with no noise (Base), at maximum structure noise (Struc.), at maximum feature noise (Feat.), the ratio of these performances (F/S) and the respective NNRD score. Both datasets report ROC-AUC. As models are fixed and evaluated over the test set, we do not invert ROC-AUC $< 0.5$.

**hiv** Starting with the GCN model, performance on the hiv is lacklustre, and NNRD indicates a very strong feature bias. Our simulated user hypothesises that the feature bias is inhibitory, and trains a GIN model, which we have shown to better utilise structural information when it is useful. Performance is greatly improved with the GIN model, but NNRD values still indicate a feature bias, so our user trains another GIN model with positional encodings included (GIN+pos). Performance decreases, but NNRD indicates near-parity between feature and structural information, and our user concludes that they have over-biased their design towards structural information. A GCN model with positional encodings (GCN+pos) confirms this; it matches the performance of the GIN without positional encodings, and has near-zero NNRD, indicating that a good balance has been found between feature and structure information.

**tox21** Our simulated user finds a strong feature bias using NNRD on the tox21 dataset. Again they attempt to alleviate this bias, training a GIN model, but find that although performance does improve somewhat, the feature bias is essentially the same as with the GCN. Given the GIN's capacity to weight towards structural information only when it is more useful, this indicates that while structural expressivity does improve performance, feature information should be prioritised for the task. Our user therefore trains a GAT, which as we've shown employs feature information with greater expressivity than structural, and includes positional encodings (GAT+pos). Here they achieve much stronger performance, and a balanced NNRD score, showing that structural information is most useful for the task when it can be explicitly combined with feature information.

| Dataset | Model | Performance | NNRD |
|---|---|---|---|
| hiv | GCN | 0.62 | −0.27 |
| | GIN | 0.74 | −0.18 |
| | GIN+pos | 0.7 | −0.075 |
| | GCN+pos | 0.74 | −0.0037 |
| tox21 | GCN | 0.69 | −0.23 |
| | GIN | 0.71 | −0.22 |
| | GAT+pos | 0.74 | −0.094 |

Table 4: Results for our simulated user on the hiv and tox21 datasets. We report ROC-AUC and NNRD.

# 6 Discussion

In our experiments we use three common GNN layers. As shown in Xu et al. (2019) the GIN's parametrisation after aggregation allows greater structural expressivity than parametrisation-first models like the GCN and GAT. We show that the all three layers are capable of graph-less learning, and that due to the parametrisation-second design, GINs are capable of feature-less learning while the others are not. While this corroborates the arguments presented in Xu et al. (2019), our work here shows that regardless of parametrisation scheme, all three message-passing layers are biased towards relying on feature information over structure. The architectural source of these behavioural differences is informative. The GCN's single shared weight matrix entangles features and structure during message-passing, providing no clean mechanism to suppress either source. For the GAT, these results are consistent with architectural design, given that the attention mechanism is feature-driven. When features are uninformative, the attention scores have no useful signal to condition on, explaining the GAT's failure on the Structure and Coupled datasets. The GIN's post-aggregation MLP cleanly separates structural summation from learned transformation, enabling it to suppress feature information when structure is more informative. Further detail on each layer's architecture is provided in Appendix A. Our results also show that the GAT layer does not integrate structural information when it is 'separate' from feature, as with our Structure and Coupled datasets. This is likely due to the attention mechanism implemented in the GAT, compared to the GCN, which means that structural information is included only in the context of feature information.

Our application of Noise-Noise Analysis over the Enzymes and Proteins datasets reproduced and extended the findings of Errica et al. (2022) and Bechler-Speicher et al. (2024). While without positional encodings, NNRD values do show structure-less learning or overfitting on structure for all layers on these datasets, with positional encodings included, all layers show both more balanced use of features and structure and have performance increases. This somewhat disagrees with the assumption made by Bechler-Speicher et al. (2024) about these tasks. Bechler-Speicher et al. (2024) argue that as Errica et al. (2022) found their highest performance with empty graphs, structure is not informative. Instead we have shown that structure is explicitly useful – all models have better performance with positional encodings included – but that these common graph layers do not usefully extract it.

The most intuitive use-case for Noise-Noise Analysis and NNRD is when expert knowledge is available. If an expert has prior knowledge that, for example, structural information is key for a given task, their model's reliance on structural information can be measured using NNRD. When the sources of useful information are not known NNRD can still be used in experimentation, as we have shown with our modelled user. This requires training at least two models, to assess if performance increases with different NNRD scores.

# 7 Conclusion

In this work we have proposed Noise-Noise Analysis, a method to measure the balance of feature and structure information used by graph-learners, along with a metric NNRD. The ability to quickly measure this balance provides a foundation for more principled model selection. We apply Noise-Noise Analysis over a synthetic and real world datasets, using the common GCN, GAT and GIN message-passing layers.

Our synthetic datasets are tailored to require use of structural or feature information, or some explicit combination of the two. We find that all layers exhibit a bias towards using feature information over structural, and that although all three are capable of graph-less learning, only the GIN performs feature-less learning. The GAT layer cannot perform tasks only over structural information, or indeed over tasks where structural and feature information must be explicitly combined. Practitioners working with datasets with strong structural weighting should strongly consider GIN-style architectures for their superior feature-less learning capabilities. We also show that structure is in fact useful on the Enzymes and Proteins datasets, but that common layers do not make use of it, challenging findings and assumptions from prior works.

**Future Work** The most clear area for future research is extension to a wider variety of graph learning tasks, including node and edge-level tasks, as well as over dynamic graphs. How to implement Noise-Noise Analysis for these cases will require re-designed noise functions, and in the case of non-supervised learning, more complex evaluation processes over those noise functions. A fuller treatment of how channel interdependence

affects NNRD is an open theoretical question and a natural direction for future work. Lastly, Noise-Noise Analysis could be extended to other multi-channel problems, for example, vision-language problems (Zhang et al., 2024).

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

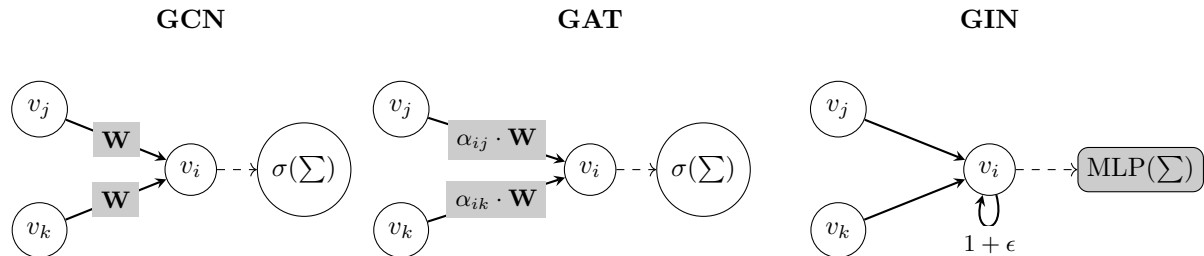

Figure 2: Parametrisation comparison of GCN, GAT, and GIN layers. GCNs employ a shared weight matrix $W$ for incoming messages, the GAT extends the GCN with learnt attention coefficients $\alpha_{ij}$, and GIN features a learnable self-loop parameter $\epsilon$ with a Multi-Layer Perceptron (MLP) after aggregation.

## A GNN Layers

**Graph Convolutional Network (GCN)**

GCNs are commonly framed as the most simple message-passing layer, with only one learnt weight matrix. For a GCN, messages are passed through a single learnt weight matrix, with the same weights applied to the states of each node. Often these messages are normalised according to the degree of the source and target of the message, so that an individual message is:

$$\text{Mess}^k(h_i^k, h_j^k) = \frac{1}{\sqrt{d_i \cdot d_j}} W^k h_j^k, \tag{7}$$

with the learnt weight matrix $W_k$ and node degrees $d_i, d_j$. Aggregation is then a simple summing over these messages, often passed through a non-linear function:

$$\text{Agg}^k[h_i^k] = \sigma(\sum_{j \in N(i)} \text{Mess}^k(h_i^k, h_j^k)). \tag{8}$$

The use of only one learnt weight matrix makes GCNs highly scalable, but the lack of parametrisation in aggregation can limit expressivity.

**Graph Isomorphism Networks (GINs)**

GINs were proposed to address the limited expressivity of GCNs and other layers, following a demonstration of the GCNs inability to express isomorphism (Xu et al., 2019). Here parametrisation is through the aggregation step, not the message-passing step, so that messages are simply:

$$\text{Mess}^k(h_i^k, h_j^k) = h_j^k. \tag{9}$$

Aggregation then makes use of an MLP or other feed-forward neural network (here we assume an NLP):

$$\text{Agg}^k[h_i^k] = \text{MLP}^k[(1 + \epsilon)h_i^k + \sum_{j \in N(i)} h_j^k]. \tag{10}$$

By avoiding averaging the node states, as the GCN does, GIN layers aim better able to express structural information. An extension is commonly applied to include edge features, termed the **GINe**, where an MLP or other feed-forward network is also applied to edge features during message-passing:

$$\text{Mess}^k(h_i^k, h_j^k) = \text{MLP}_e^k(e_{ij}) \cdot h_j^k. \tag{11}$$

Here $e_{ij}$ are the features attached to the edge between nodes $i$ and $j$. Important to note is that these features, unlike node features, are not necessarily updated each layer.

**Graph Attention (GAT)**

An assumption made by both the GCN and the GIN is that the messages from each neighbouring node are equally important. GAT layers use multiple learnt weight matrixes to allow varying weights $\alpha_{i,j}$ ('attention scores') to be applied to each incoming message, aiming to allow more important information to be given greater weight:

$$\text{Mess}^k(h_i^k, h_j^k) = \alpha_{i,j} W^k h_j^k. \tag{12}$$

As for GCNs, $W^k$ here is a learnable weight matrix. Attention scores are computed using a learnable vector $a$, averaged across all other neighbours:

$$\alpha_{i,j} = \frac{\exp(\text{LR}(a^T[W^k h_i^k || W^k h_j^k]))}{\sum_{n \in N(i)} \exp(\text{LR}(a^T[W^k h_i^k || W^k h_n^k]))}. \tag{13}$$

Here $||$ indicates concatenation and LR a leaky ReLu function. Although this requires computing all $W^k h_v^k$, which adds considerably to computational cost, it does lend the ability to focus on some neighbours more than others. This has been particularly useful in chemistry.

Aggregation is then the same process as for a GCN:

$$\text{Agg}^k[h_i^k] = \sigma\Big(\sum_{j \in N(i)} \text{Mess}^k(h_i^k, h_j^k)\Big). \tag{14}$$

The use of the same aggregation method, as well as the parametrisation with $W^k$ during message-passing, makes the GAT layer highly comparable to the GCN layer. Without emphasising structure like the GIN does, the GAT may lean more towards feature-based information than the GIN. Indeed, the emphasis on features during message-passing may lead to a greater reliance on feature information than the GCN.

## B  Theory on Noise Functions

In Section 3 we introduced specific noising functions for Noise-Noise Analysis. Here we prove these functions achieve equal proportional information loss rates, validating direct comparison of their performance curves.

### B.1  Formalization

Consider dataset $\mathcal{D} = \{(G_i, y_i)\}_{i=1}^N$ where $G_i = (V_i, E_i, X_i)$ with features $X_i \in \mathbb{R}^{|V_i| \times d}$ and labels $y_i \in \{0, 1\}$.

Our noising functions at level $t \in [0, 1]$ randomly select $tN$ graphs and replace:

**Feature noise** $\mathcal{N}_t^X$: Replace $X_i$ with $X_i^{\text{rand}}$ sampled uniformly from $[\min(X), \max(X)]$

**Structure noise** $\mathcal{N}_t^E$: Replace $E_i$ with ER graph having same edge count, **preserving features $X_i$**

This creates a mixture where proportion $(1-t)$ retain original information and proportion $t$ have noise. The decomposition in the mutual information that follows holds because the mixing is Bernoulli-indexed and independent of labels. Let $Z_i \sim \text{Bern}(t)$ indicate whether graph $i$ is noised. By the law of total expectation over $Z$:

$$I(y; \mathcal{N}_t^X(X)) = (1 - t) \cdot I(y; X) + t \cdot I(y; X^{\text{rand}}) \tag{15}$$

Since $X^{\text{rand}}$ is independent of $y$ by construction, $I(y; X^{\text{rand}}) = 0$ (Lemma 1), and the decomposition follows. The same argument holds for structure noise.

## B.2 Random Feature Noise

**Lemma 1** (Random Features). *If $X_i^{rand}$ are sampled uniformly and independently of y, then $I(y; X^{rand}) = 0$ in expectation, with finite-sample spurious correlations bounded by $O(d \log(N)/N)$.*

*Proof.* By independence, $P(y|X^{rand}) = P(y)$, so mutual information is zero. Random chance introduces spurious correlations of order $O(d \log(N)/N)$ from standard concentration results. $\square$

For typical datasets ($N \geq 1000$, $d \leq 100$), this bound is negligible. We demonstrate insensitivity to feature dimensions empirically in Section D.

## B.3 Erdös-Rényi Structure Noise

Our ER noise preserves the node and edge count within graphs, in order that all feature information can still be included post-noising. In this case, the only mutual information still accessible that might pertain to $y$ is through counting nodes and edges; we show, using results from (Xu et al., 2019), that over ER graphs even this information is minimally accessible. This proof regards information in general, not necessarily pertaining to a given task — indeed, if node and edge counting was strongly informative for a task, we can assume that a GNN would not be necessary in the first place.

We define GNN-accessible information $I_{\mathcal{F}}(y; E) := \sup_{f \in \mathcal{F}} I(y; f(E))$ as the maximum information any GNN in class $\mathcal{F}$ can extract.

**Lemma 2** (ER Graphs for GNNs). *Let $ER_i$ be ER graphs with $n_i$ nodes and $m_i$ edges, generated independently of $y_i$, with original features $X_i$ preserved. Then $I_{\mathcal{F}}(y; E_{ER}|X) = O(\log(N)/N)$.*

*Proof.* (Xu et al., 2019) showed that $k$-layer message-passing GNNs compute functions of the $k$-WL graph coloring. ER graphs with uniform node features are approximately regular, with all nodes having similar degree $2m/n$. Therefore the $k$-WL coloring assigns similar colors to all nodes when $k$ is small relative to the graph diameter $\approx \log(n)$. Even if graph sizes vary, (Xu et al., 2019) further showed that $k$-layer GNNs cannot count nodes globally when $k <$ diameter. Since typical GNNs use $k = 3\text{-}10 \ll \log(n)$ for $n \geq 20$, size information is inaccessible.

Since $f(ER_i)$ is approximately constant, the mutual information $I(y; f(E_{ER})) \approx 0$, with finite-sample bound $O(\log(N)/N)$. $\square$

## B.4 Main Result

**Theorem 3** (Equal Proportional Loss). *For datasets where $I_{\mathcal{F}}(y; X), I_{\mathcal{F}}(y; E) \gg O(\log(N)/N)$, our noising functions achieve equal proportional information loss rates of $-1$, validating direct comparison of performance curves in NNRD (Equation 5).*

*Proof.* From the mixture decomposition and Lemmas 1-2:

$$I_{\mathcal{F}}(y; \mathcal{N}_t^X(X)) \approx (1-t) \cdot I_{\mathcal{F}}(y; X) \tag{16}$$

$$I_{\mathcal{F}}(y; \mathcal{N}_t^E(E)) \approx (1-t) \cdot I_{\mathcal{F}}(y; E) \tag{17}$$

Taking derivatives: $\frac{\partial I_{\mathcal{F}}}{\partial t} \approx -I_{\mathcal{F}}$ for both sources. The proportional rates are therefore both equal to $-1$, with error $O(\epsilon/I_{\mathcal{F}})$ that is negligible for informative datasets. $\square$

This theoretical foundation justifies our core assumption (Equation 6) and explains why all-or-nothing replacement is appropriate compared to in-sample perturbations which do not satisfy these criteria.

## C   Rough Bounds on NNRD

For our first two examples, we assume that performance at $t = 0$ is perfect, and $h$ is a performance metric where $h = 1$ is perfect and $h = 0.5$ is random guessing. Next, consider a dataset where all useful information comes from either features or structure. With a linear response to the loss in information from the noising function, from $h = 1$ to $h = 0.5$, we have:

$$\text{NNRD} = \int_0^{T=1} \log(1 - t/2)dt = -0.307. \tag{18}$$

We can also model the results found by Bechler-Speicher et al. (2024), in which a model overfits on structure, to the detriment of performance. This implicates that performance is initially not ideal, but might rise to be ideal with increasing structural noise, and feature noise only degrades performance. In this case NNRD exceeds the 0.307 bound, as observed in Table 3 over the Enzymes and Proteins datasets.

A benefit of NNRD is that it is symmetric; with the same performance curves reversed between structure and features, the same NNRD values would be returned with positive signs.

## D   Sensitivity Analysis

| Dataset | Slope ($\times 10^{-4}$) | $R^2$ | Sprmn. $\rho$ | $p$-value |
|---|---|---|---|---|
| Easy | 2.26 | 0.244 | 0.543 | **0.006** |
| Structure | 5.40 | 0.070 | -0.017 | 0.936 |
| Feature | -0.98 | 0.041 | -0.222 | 0.298 |
| Coupled | 4.28 | 0.092 | 0.120 | 0.576 |

Table 5: Correlation analysis between timestep size and CV(NNRD) across synthetic datasets. Slope values are scaled by $10^{-4}$ for readability. Bold p-values indicate statistical significance ($p < 0.05$). Only synth-easy shows significant correlation between timestep size and NNRD variability, with all other experiments showing no significant relationship.

We perform a sensitivity analysis for NNRD over timestep size, ie $|T|$, $T = \{t_0, t_1, \ldots\}$, $t \in [0, 1]$. Over our four synthetic datasets we sample $|T| \in [4, 96]$, with uniform timestep sizes, then train a model and evaluate NNRD five times for each $|T|$. Other experimental settings are identical to those in the main body of the paper.

We measure variation in critical values $CV(\text{NNRD}) = \text{dev}(\text{NNRD})/\bar{\text{NNRD}}$ with timestep size to evaluate NNRD's stability. Table 5 shows the results of this analysis. NNRD calculations remain stable across different timestep sizes, with coefficient of variation values ranging from $CV = -0.057$ to $CV = 0.080$. Linear trend analysis shows negligible slopes across all conditions, with the largest absolute slope being $|\beta| = 0.000281$ for synth-easy, indicating minimal systematic drift with timestep size. The synth-coupled condition, representing the most complex balanced information scenario, shows near-zero slope ($\beta = 0.000044$) with very low explanatory power ($R^2 = 0.011$).

Only one of four experimental conditions shows significant correlation between timestep size and CV(NNRD), with synth-easy yielding $r = 0.494$ and $p = 0.014$, but with minimal slope. The remaining three conditions show non-significant correlations (all $p > 0.05$), with synth-structure ($p = 0.212$), synth-feature ($p = 0.345$), and synth-coupled ($p = 0.150$) demonstrating independence from timestep size. The overall finding that only 25% of experiments show significant timestep-CV correlation confirms that NNRD measurements are robust to temporal discretization choices.

## E   Consistency & Invariance

As discussed above, the difficult to treat setting where we have neither model nor dataset specified makes theoretically verifying key properties of NNRD (consistency, invariance) unfeasible. Instead we evaluated the

NNRD metric using our synthetic datasets, given their known information balance, varying several 'nuisance parameters' to demonstrate the stability and consistency of NNRD. These nuisance parameters should be orthogonal to the given task:

**Feature Dim.** For node and edge features $X \in \mathbb{R}^d$, vary $d$.

**Node Count** For graphs of node count $|V|$, vary $|V|$.

**Density** The edge-count $|E|$ in our ER graphs. This could pertain to the structural signal, so we exclude the Structure-Density dataset and nuisance parameter combination from our experiments.

Each dataset contains 1,600 graphs split into training (70%), validation (20%), and test (10%) sets. For all experiments, we employ a Graph Isomorphism Network (GIN) with three layers, a hidden dimension of 100, and train for 25 epochs using the Adam optimiser with a learning rate of 0.001. For the feature dimensionality experiments, we generate datasets with 2, 5, 10, 25, and 50 feature dimensions for both node and edge attributes. The graph size experiments use four node count ranges: 20-50, 50-100, 100-200, and 200-400 nodes per graph. The density experiments, conducted exclusively on our feature-labelled synthetic datasets, test edge densities of 0.05, 0.1, 0.2, 0.3, and 0.5. We exclude structure-labelled data from density experiments as varying edge density fundamentally alters the signal between circular ladder and random graph classes. Each experimental configuration was repeated three times with freshly generated datasets to account for sampling variance. The results of this testing can be found in Table 2 (in the main text), and visualised in Figure E.

Through Kruskal-Wallis testing over NNRD values, and Levene testing over variance, these results show that NNRD is both consistent and invariant to irrelevant nuisance parameters.

## F  Complete Results

Here we present the complete results produced and discussed in this work. In Figures (4, 5, 6) we provide visualisations of Noise-Noise curves on our synthetic datasets for the GCN, GAT and GIN respectively. In Figure 7 we include a separate visualisation for the GCN on the Structure dataset, using accuracy instead of ROC-AUC, to explain the U-shaped curve displayed in Figure 4. In Figure 8 we present results for all three layers on the Enzymes and Proteins datasets, along with the same with positional encodings included. Finally, in Tables (6, 7, 8) we present results for all datasets for the GCN, GAT and GIN respectively.

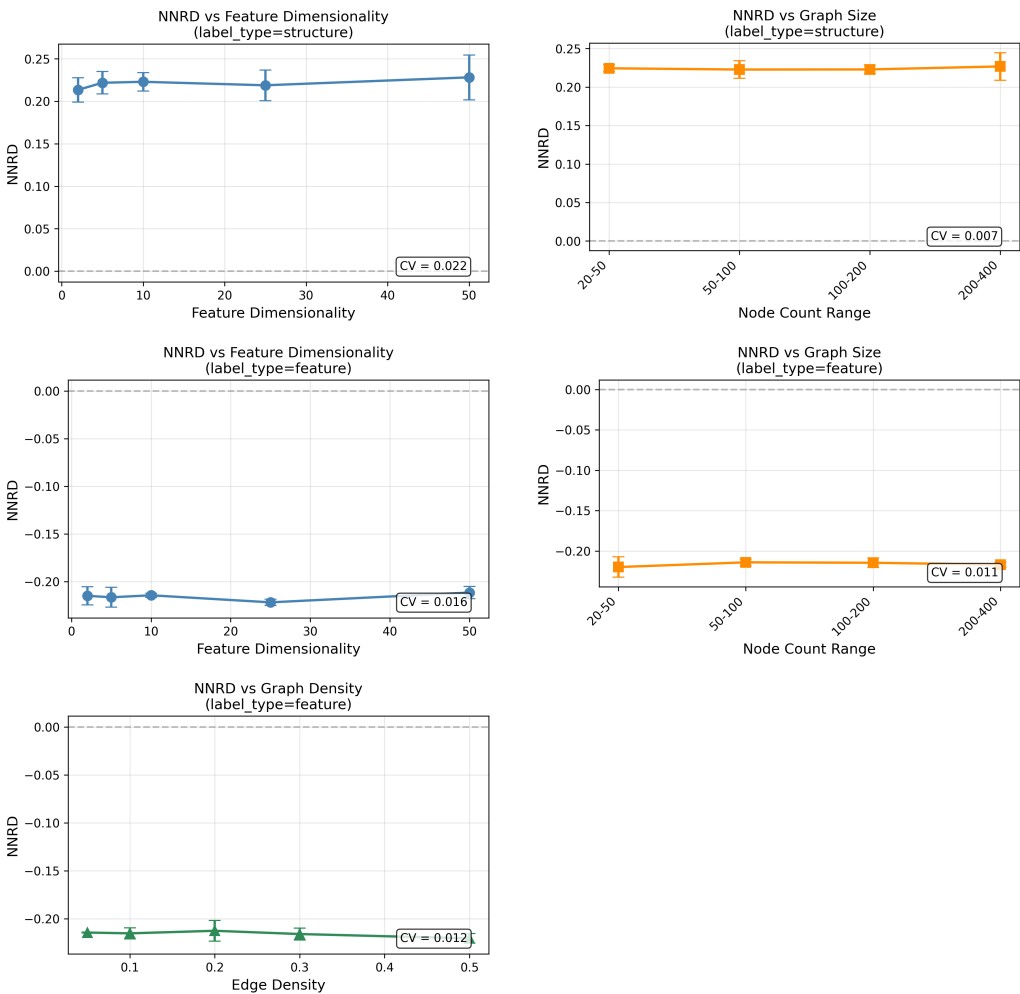

Figure 3: NNRD over varied nuisance parameters for our structure and feature datasets.

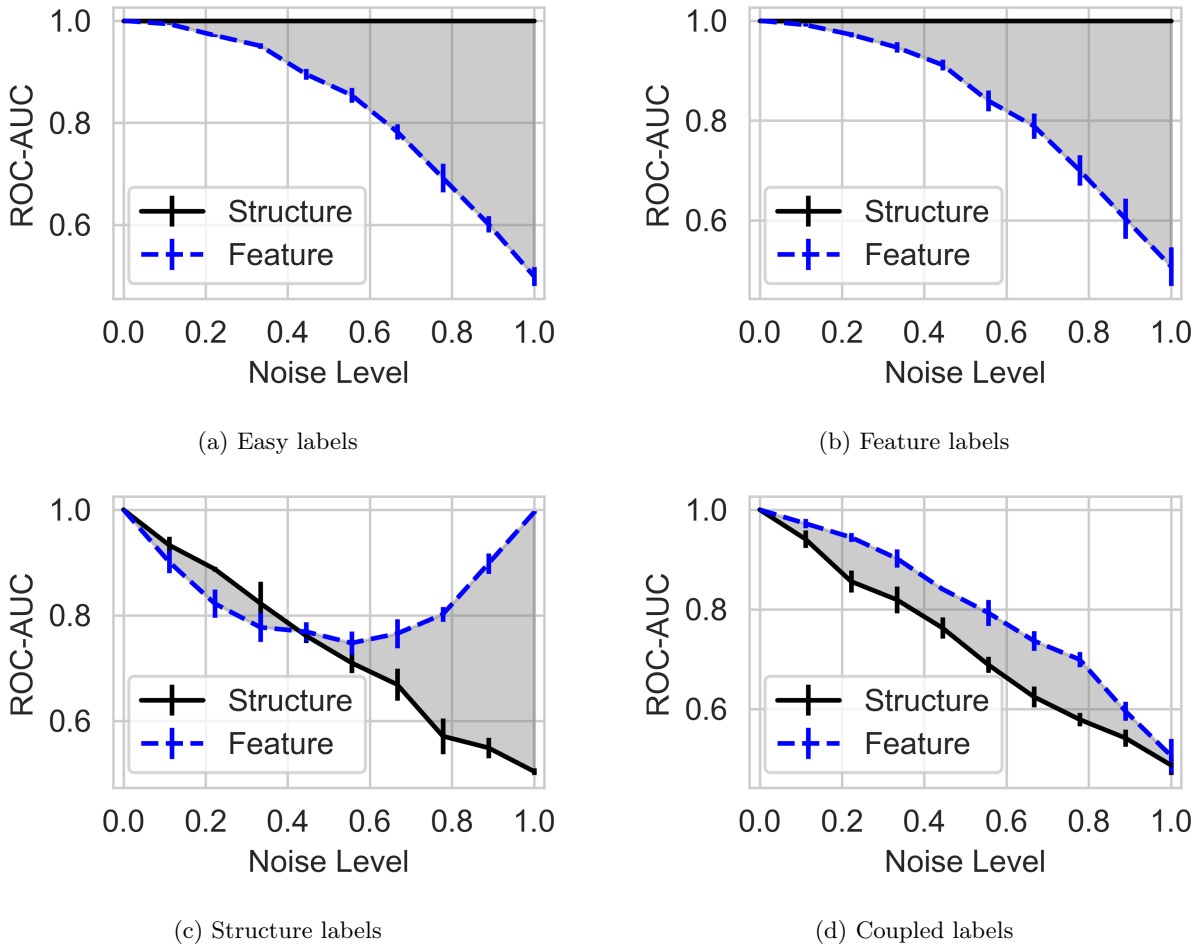

(a) Easy labels

(b) Feature labels

(c) Structure labels

(d) Coupled labels

Figure 4: Performance variation for supervised training of our GCN models over synthetic datasets with increasing noise on structure and features in the test-set. All datasets report ROC-AUC. The GCN relies on feature information exclusively on the Easy dataset and easily ignores structure on the Feature dataset. On the Structure dataset there is a drift in probabilities as feature noise increases, leading to a U-shaped curve. On the Coupled dataset features and structures are equally used.

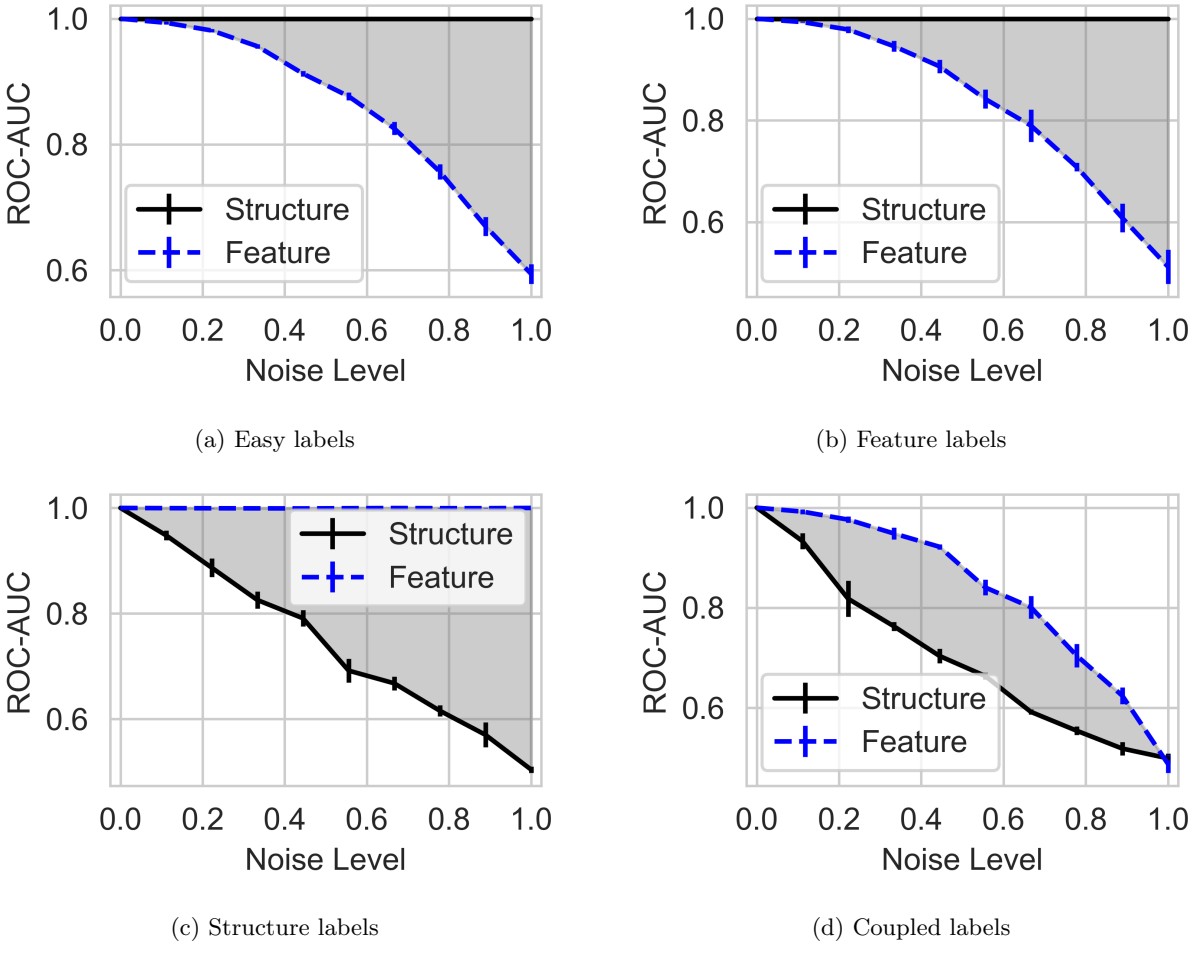

(a) Easy labels

(b) Feature labels

(c) Structure labels

(d) Coupled labels

Figure 5: Performance variation for supervised training of our GIN models over synthetic datasets with increasing noise on structure and features in the test-set. All datasets report ROC-AUC. The GIN relies on feature information exclusively on the Easy dataset and easily ignores structure on the Feature dataset. The inverse is true on the structure dataset. On the Coupled dataset features and structures are equally used.

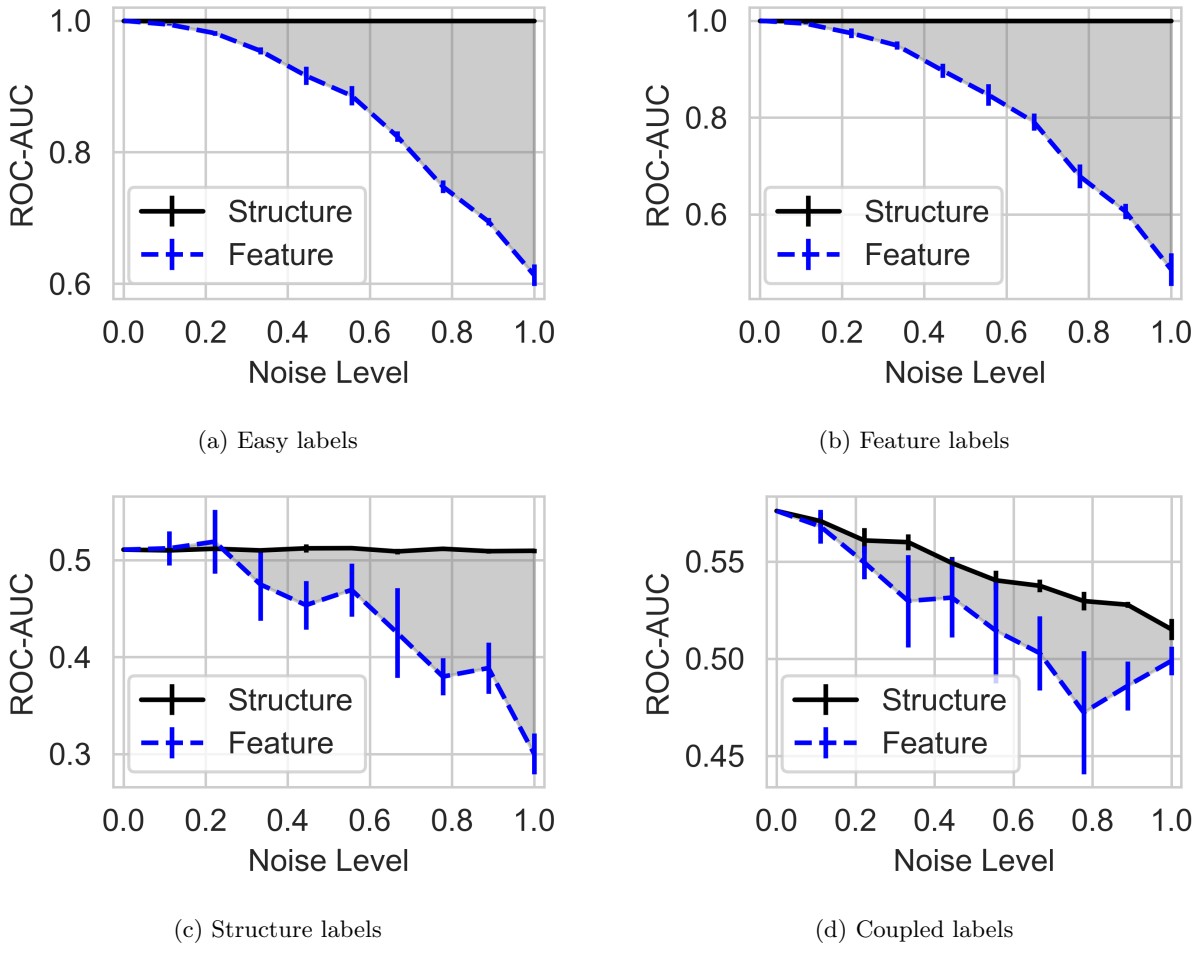

(a) Easy labels

(b) Feature labels

(c) Structure labels

(d) Coupled labels

Figure 6: Performance variation for supervised training of our GAT models over synthetic datasets with increasing noise on structure and features in the test-set. All datasets report ROC-AUC. The GAT relies on feature information exclusively on the Easy dataset and easily ignores structure on the Feature dataset. On the Structure dataset performance is never better than random, with decreasing performance as feature noise increases. Performance on the Coupled dataset performance is poor, but performance does decrease over both feature and structure noise.

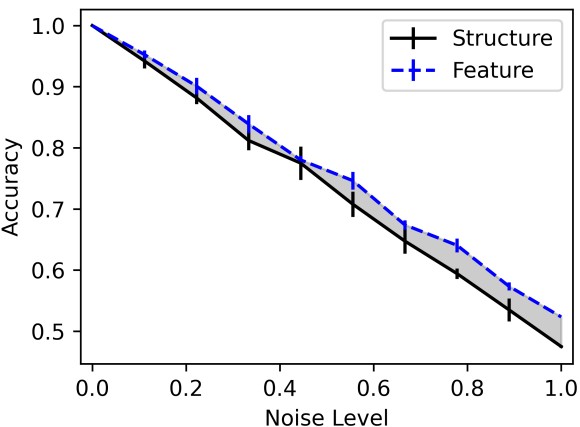

Figure 7: Accuracy for a GCN model on the Structure dataset with increasing structure and feature noise.

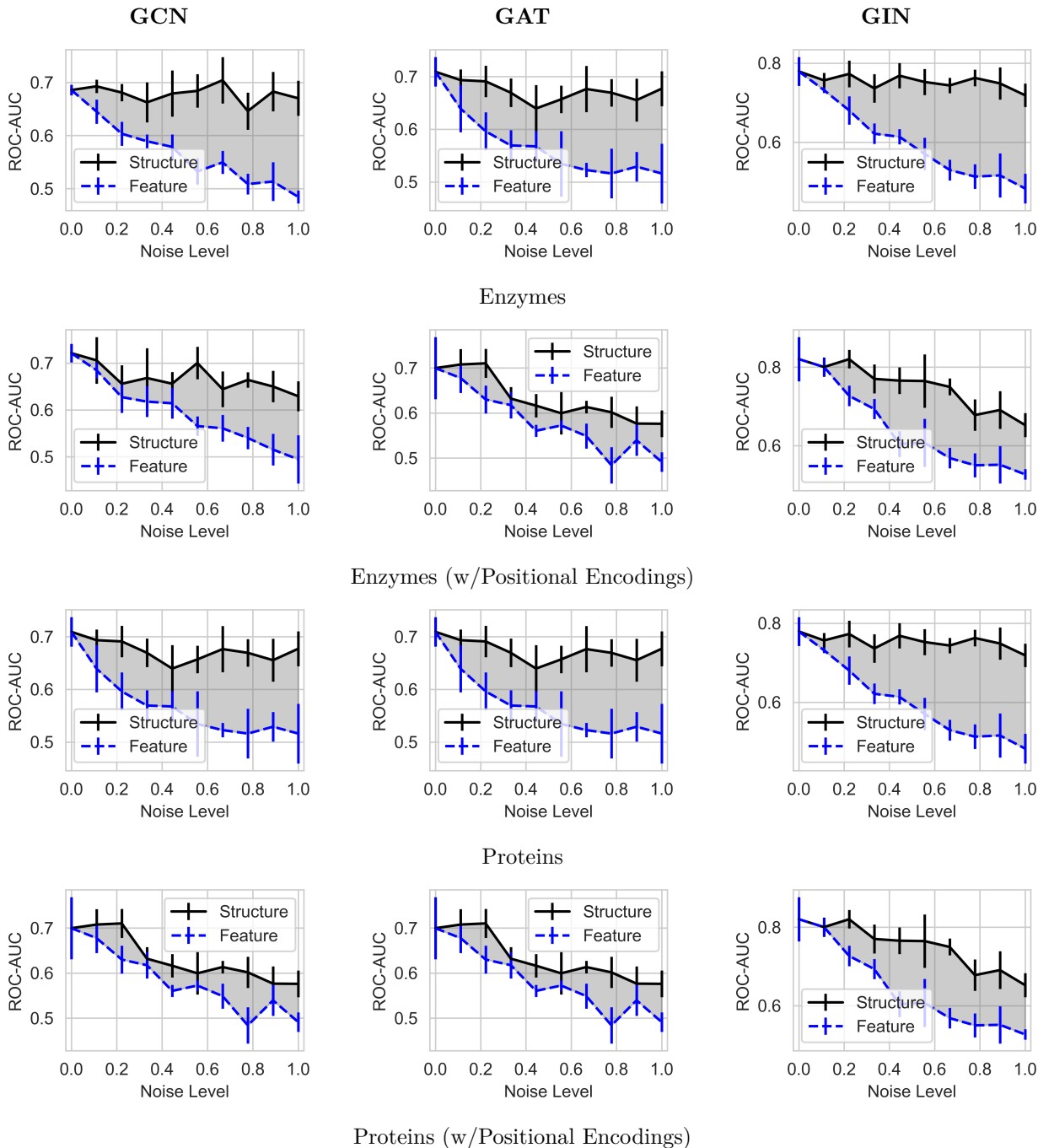

Figure 8: Performance variation for supervised training of our GCN, GAT and GIN models over the Proteins and Enzymes datasets with increasing noise on structure and features in the test-set. All datasets report ROC-AUC.

| Dataset | Pos. Enc. | Base | Struc. | Feat. | F/S | NNRD |
|---------|-----------|------|--------|-------|-----|------|
| Enzymes | False | 0.69±0.00 | 0.67±0.03 | 0.48±0.01 | 0.72 | −0.33 |
| Enzymes | True | 0.72±0.02 | 0.63±0.03 | 0.49±0.05 | 0.78 | −0.23 |
| Proteins | False | 0.72±0.03 | 0.74±0.05 | 0.49±0.05 | 0.66 | −0.37 |
| Proteins | True | 0.78±0.04 | 0.75±0.03 | 0.6±0.0 | 0.8 | −0.28 |
| molbace | False | 0.58±0 | 0.58±0.04 | 0.45±0.14 | 0.78 | −0.13 |
| molbace | True | 0.79±0 | 0.73±0.06 | 0.75±0.02 | 1.0 | −0.075 |
| molbbbp | False | 0.64±0 | 0.59±0.02 | 0.48±0.03 | 0.81 | −0.18 |
| molbbbp | True | 0.71±0 | 0.55±0.03 | 0.55±0.01 | 1.0 | −0.023 |
| molclintox | False | 0.47±0 | 0.55±0.03 | 0.46±0.13 | 0.84 | −0.066 |
| molclintox | True | 0.67±0 | 0.57±0.03 | 0.52±0.05 | 0.91 | −0.12 |
| molhiv | False | 0.62±0 | 0.56±0.02 | 0.51±0.02 | 0.91 | −0.27 |
| molhiv | True | 0.74±0 | 0.53±0.01 | 0.55±0.02 | 1.0 | −0.0037 |
| molsider | False | 0.54±0 | 0.56±0.01 | 0.51±0.01 | 0.91 | −0.15 |
| molsider | True | 0.54±0 | 0.49±0.03 | 0.52±0.01 | 1.1 | 0.0023 |
| moltox21 | False | 0.69±0 | 0.64±0.00 | 0.49±0.01 | 0.77 | −0.23 |
| moltox21 | True | 0.71±0 | 0.54±0.00 | 0.49±0.01 | 0.91 | −0.12 |
| easy | False | 1.0±0 | 1.0±0 | 0.5±0.0 | 0.5 | −0.22 |
| easy | True | 1.0±0 | 1.0±0 | 0.98±0.00 | 0.98 | −0.27 |
| feature | False | 1.0±0 | 1.0±0 | 0.51±0.03 | 0.51 | −0.22 |
| feature | True | 1.0±0 | 1.0±0 | 0.53±0.02 | 0.53 | −0.23 |
| structure | False | 1.0±0 | 0.5±0.0 | 1.0±0.0 | 2.0 | 0.16 |
| structure | True | 1.0±0 | 0.5±0.0 | 1.0±0 | 2.0 | 0.31 |
| coupled | False | 1.0±0 | 0.49±0.01 | 0.51±0.03 | 1.0 | 0.092 |
| coupled | True | 1.0±0 | 0.5±0.0 | 0.45±0.01 | 0.9 | 0.071 |

Table 6: Results for a trained model over the test-set of our datasets for a GCN model. We report whether the model uses positional encodings, the base performance with un-noised test sets, the performance at maximum feature and structure noise, the ratio between these fully noised performances, and the NNRD score.

# G  Node-Level Experiments

Our method requires whole-graph noise functions, which restricts it to graph-level tasks and to the special case where all node labels in a graph are predicted simultaneously. Full node and edge-level extension is not tractable: within-sample noise functions are not well-justified (Section 3), and designing valid synthetic datasets at node level is challenging because node classification is essentially predicting an additional node feature, making full isolation of information sources difficult. Extension to edge-level tasks is not tractable since it is unclear which edges *should* be present when all edges are random. Most node-level tasks are also excluded due to reliance on labels of neighbouring nodes.

That said, graph-level tasks share the same MPNN mechanism as node and edge-level tasks (differing only in the final pooling step), so our findings on graph-level tasks transfer appropriately. Within the limited case of predicting all node labels simultaneously, we developed four synthetic node-classification datasets with controlled isolation of information sources, directly analogous to our graph-level ones.

**Node-Level Synthetic Datasets**

Each node is assigned a binary label $y_i \in \{0, 1\}$. Datasets are constructed as follows, with labelling schemes visualised in Figure 10.

**Easy.** Both information sources are predictive. Each node is assigned a binary label $y_i \in \{0, 1\}$ uniformly at random. The node feature $x_i \in \{0.25, 0.5\}$ encodes $y_i$ directly, and class-1 nodes are wired into

| Dataset | Pos. Enc. | Base | Struc. | Feat. | F/S | NNRD |
|---|---|---|---|---|---|---|
| Enzymes | False | 0.78±0.03 | 0.72±0.02 | 0.48±0.03 | 0.67 | −0.33 |
| Enzymes | True | 0.82±0.05 | 0.65±0.03 | 0.53±0.01 | 0.82 | −0.25 |
| Proteins | False | 0.76±0.04 | 0.81±0.04 | 0.4±0.0 | 0.49 | −0.31 |
| Proteins | True | 0.82±0.02 | 0.71±0.04 | 0.51±0.12 | 0.72 | −0.26 |
| molbace | False | 0.42±0 | 0.67±0.09 | 0.51±0.15 | 0.76 | −0.16 |
| molbace | True | 0.51±0 | 0.65±0.1 | 0.39±0.15 | 0.6 | −0.28 |
| molbbbp | False | 0.66±0 | 0.57±0.01 | 0.48±0.03 | 0.84 | −0.26 |
| molbbbp | True | 0.67±0 | 0.53±0.03 | 0.54±0.04 | 1.0 | 0.065 |
| molclintox | False | 0.48±0 | 0.47±0.05 | 0.49±0.04 | 1.0 | 0.081 |
| molclintox | True | 0.66±0 | 0.52±0.03 | 0.51±0.02 | 0.98 | 0.003 |
| molhiv | False | 0.74±0 | 0.55±0.01 | 0.51±0.04 | 0.93 | −0.18 |
| molhiv | True | 0.7±0 | 0.51±0.01 | 0.5±0.0 | 0.98 | −0.075 |
| molsider | False | 0.57±0 | 0.54±0.01 | 0.51±0.03 | 0.94 | −0.18 |
| molsider | True | 0.55±0 | 0.51±0.01 | 0.52±0.01 | 1.0 | 0.17 |
| moltox21 | False | 0.71±0 | 0.62±0.01 | 0.48±0.00 | 0.77 | −0.22 |
| moltox21 | True | 0.73±0 | 0.56±0.02 | 0.49±0.00 | 0.87 | −0.13 |
| easy | False | 1.0±0 | 1.0±0 | 0.59±0.01 | 0.59 | −0.22 |
| easy | True | 1.0±0 | 1.0±0 | 0.98±0.00 | 0.98 | −0.2 |
| feature | False | 1.0±0 | 1.0±0 | 0.51±0.03 | 0.51 | −0.22 |
| feature | True | 1.0±0 | 1.0±0 | 0.54±0.02 | 0.54 | −0.23 |
| structure | False | 1.0±0 | 0.5±0.0 | 1.0±0 | 2.0 | 0.31 |
| structure | True | 1.0±0 | 0.5±0.0 | 1.0±0 | 2.0 | 0.31 |
| coupled | False | 1.0±0 | 0.5±0.0 | 0.49±0.01 | 0.98 | 0.16 |
| coupled | True | 1.0±0 | 0.49±0.01 | 0.46±0.00 | 0.94 | 0.039 |

Table 7: Results for a trained model over the test-set of our datasets for a GIN model. We report whether the model uses positional encodings, the base performance with un-noised test sets, the performance at maximum feature and structure noise, the ratio between these fully noised performances, and the NNRD score.

triangles (3-cliques) while class-0 nodes form a single cycle, so triangle membership also encodes $y_i$. All nodes have degree 2, ensuring that degree alone carries no signal.

**Feature-only.** Only node features are predictive. Labels and features are assigned as in **Easy**, but triangle membership is randomised independently of $y_i$, so local graph structure is uninformative.

**Structure-only.** Only graph structure is predictive. Triangle membership encodes $y_i$ as in **Easy**, but node features are drawn independently of the label, rendering them uninformative.

**Coupled.** Neither source is individually sufficient. Two independent balanced binary variables $a_i, b_i \sim$ Bern(0.5) are drawn per node. The label is $y_i = \mathbf{1}[a_i = b_i]$; the feature encodes $a_i$ and triangle membership encodes $b_i$. Predicting $y_i$ therefore requires integrating both sources: neither feature nor structure alone is more than chance-level informative.

**Results**

These experiments are presented as a first step towards full extension to node and edge-level tasks; findings should be taken primarily from our graph-level results. Quantitative results are presented in Table 9 and visualised in Figure 9. Results broadly mirror those of our graph-level tasks, though the three layers more evenly balance the use of feature and structure information on the Easy dataset. The GAT model heavily overfits on feature information on the Structure dataset, and no model achieves perfect performance even under zero noise on the Coupled dataset.

| Dataset | Pos. Enc. | Base | Struc. | Feat. | F/S | NNRD |
|---------|-----------|------|--------|-------|-----|------|
| Enzymes | False | 0.71±0.02 | 0.68±0.03 | 0.52±0.05 | 0.76 | −0.38 |
| Enzymes | True | 0.7±0.0 | 0.58±0.03 | 0.49±0.02 | 0.84 | −0.16 |
| Proteins | False | 0.7±0.0 | 0.69±0.03 | 0.51±0.04 | 0.74 | −0.31 |
| Proteins | True | 0.8±0.0 | 0.75±0.03 | 0.67±0.07 | 0.89 | −0.16 |
| molbace | False | 0.46±0 | 0.49±0.12 | 0.52±0.14 | 1.1 | 0.035 |
| molbace | True | 0.57±0 | 0.75±0.03 | 0.59±0.07 | 0.79 | −0.092 |
| molbbbp | False | 0.65±0 | 0.62±0.01 | 0.46±0.04 | 0.74 | −0.31 |
| molbbbp | True | 0.7±0 | 0.54±0.01 | 0.54±0.02 | 1.0 | −0.066 |
| molclintox | False | 0.51±0 | 0.47±0.07 | 0.58±0.12 | 1.2 | 0.11 |
| molclintox | True | 0.66±0 | 0.44±0.12 | 0.41±0.09 | 0.93 | −0.034 |
| molhiv | False | 0.67±0 | 0.57±0.01 | 0.48±0.02 | 0.84 | −0.22 |
| molhiv | True | 0.69±0 | 0.6±0.0 | 0.47±0.02 | 0.78 | −0.14 |
| molsider | False | 0.57±0 | 0.57±0.01 | 0.5±0.0 | 0.88 | −0.26 |
| molsider | True | 0.53±0 | 0.53±0.02 | 0.53±0.03 | 1.0 | −0.19 |
| moltox21 | False | 0.71±0 | 0.67±0.01 | 0.5±0.0 | 0.75 | −0.24 |
| moltox21 | True | 0.74±0 | 0.59±0.00 | 0.51±0.01 | 0.86 | −0.094 |
| easy | False | 1.0±0 | 1.0±0 | 0.61±0.01 | 0.61 | −0.22 |
| easy | True | 1.0±0 | 1.0±0 | 0.96±0.00 | 0.96 | −0.25 |
| feature | False | 1.0±0 | 1.0±0 | 0.49±0.03 | 0.49 | −0.21 |
| feature | True | 1.0±0 | 1.0±0 | 0.46±0.02 | 0.46 | −0.21 |
| structure | False | 0.51±0 | 0.51±0.00 | 0.3±0.0 | 0.59 | −0.19 |
| structure | True | 1.0±0 | 0.5±0.0 | 1.0±0 | 2.0 | 0.31 |
| coupled | False | 0.58±0 | 0.52±0.00 | 0.5±0.0 | 0.96 | −0.16 |
| coupled | True | 0.97±0 | 0.5±0.0 | 0.5±0.0 | 1.0 | 0.039 |

Table 8: Results for a trained model over the test-set of our datasets for a GAT model. We report whether the model uses positional encodings, the base performance with un-noised test sets, the performance at maximum feature and structure noise, the ratio between these fully noised performances, and the NNRD score.

| Layer | Dataset | Base | Structure | Feature | F/S | NNRD |
|-------|---------|------|-----------|---------|-----|------|
| GCN | Easy | 1.00±0.00 | 0.65±0.00 | 0.5±0.0 | 0.77 | −0.063 |
|     | Feature | 0.82±0.01 | 0.65±0.00 | 0.5±0.0 | 0.77 | −0.13 |
|     | Structure | 0.96±0.00 | 0.5±0.0 | 0.75±0.00 | 1.5 | 0.24 |
|     | Coupled | 0.78±0.01 | 0.5±0.0 | 0.5±0.0 | 1.0 | 0.052 |
| GIN | Easy | 1.00±0.00 | 0.63±0.00 | 0.5±0.0 | 0.79 | −0.069 |
|     | Feature | 1.00±0.00 | 0.98±0.00 | 0.5±0.0 | 0.51 | −0.22 |
|     | Structure | 0.96±0.00 | 0.5±0.0 | 0.73±0.00 | 1.5 | 0.20 |
|     | Coupled | 0.96±0.00 | 0.5±0.0 | 0.5±0.0 | 1.0 | 0.10 |
| GAT | Easy | 1.00± 0.00 | 0.66±0.00 | 0.49±0.00 | 0.74 | −0.084 |
|     | Feature | 1.00± 0.00 | 0.97±0.00 | 0.5±0.0 | 0.52 | −0.29 |
|     | Structure | 0.66±0.07 | 0.5±0.0 | 0.85±0.00 | 1.7 | 0.31 |
|     | Coupled | 0.84±0.07 | 0.5±0.0 | 0.5±0.0 | 1.0 | 0.017 |

Table 9: Performance for GCN, GIN and GAT models on the synthetic node datasets. We report the performance with no noise (Base), at maximum structure noise (Structure), at maximum feature noise (Feature), the ratio of these performances (F/S) and the respective NNRD score.

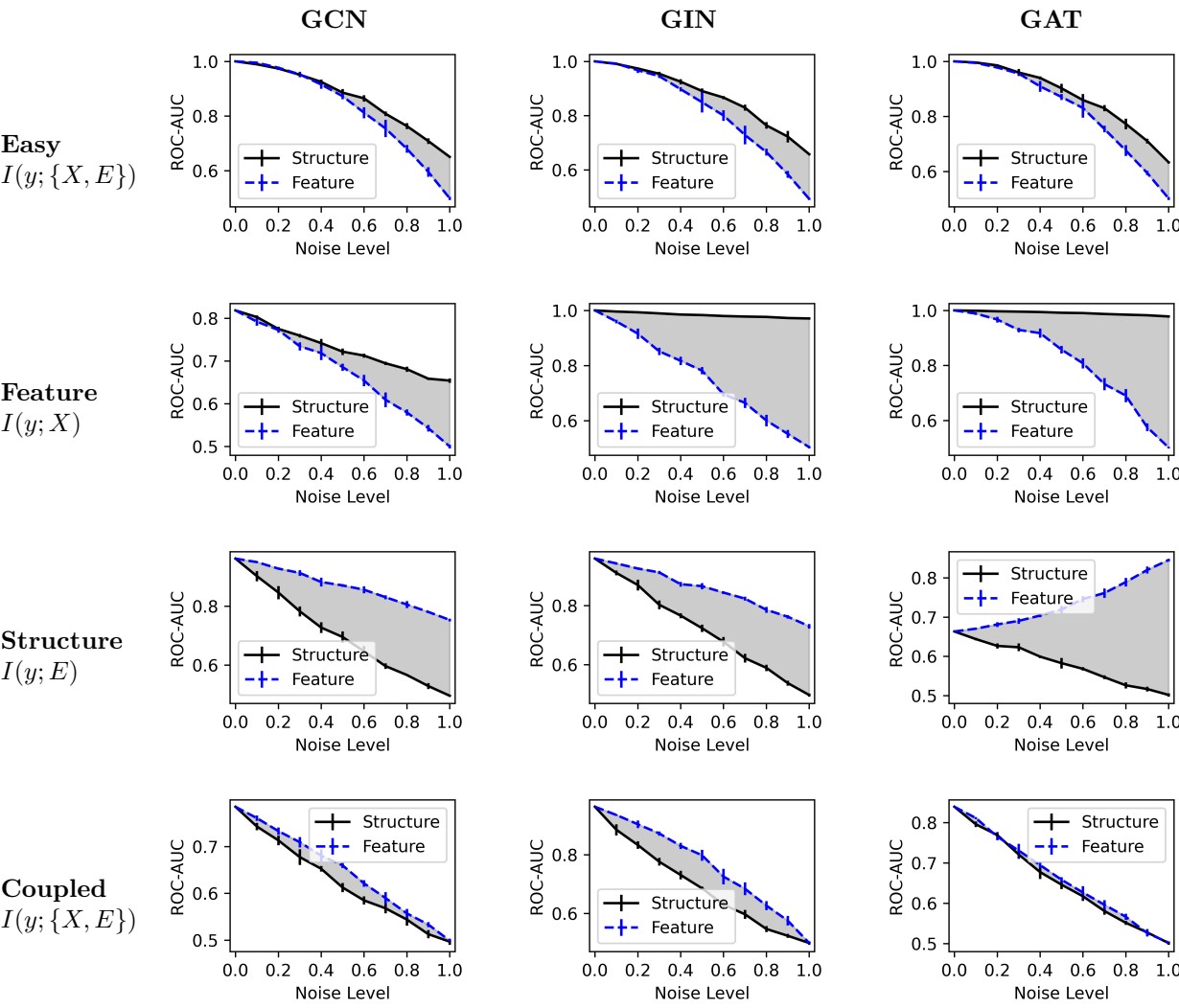

Figure 9: Performance variation for supervised training of our GCN, GIN and GAT models over node classification synthetic datasets with increasing noise on structure and features in the test-set. All datasets report ROC-AUC.

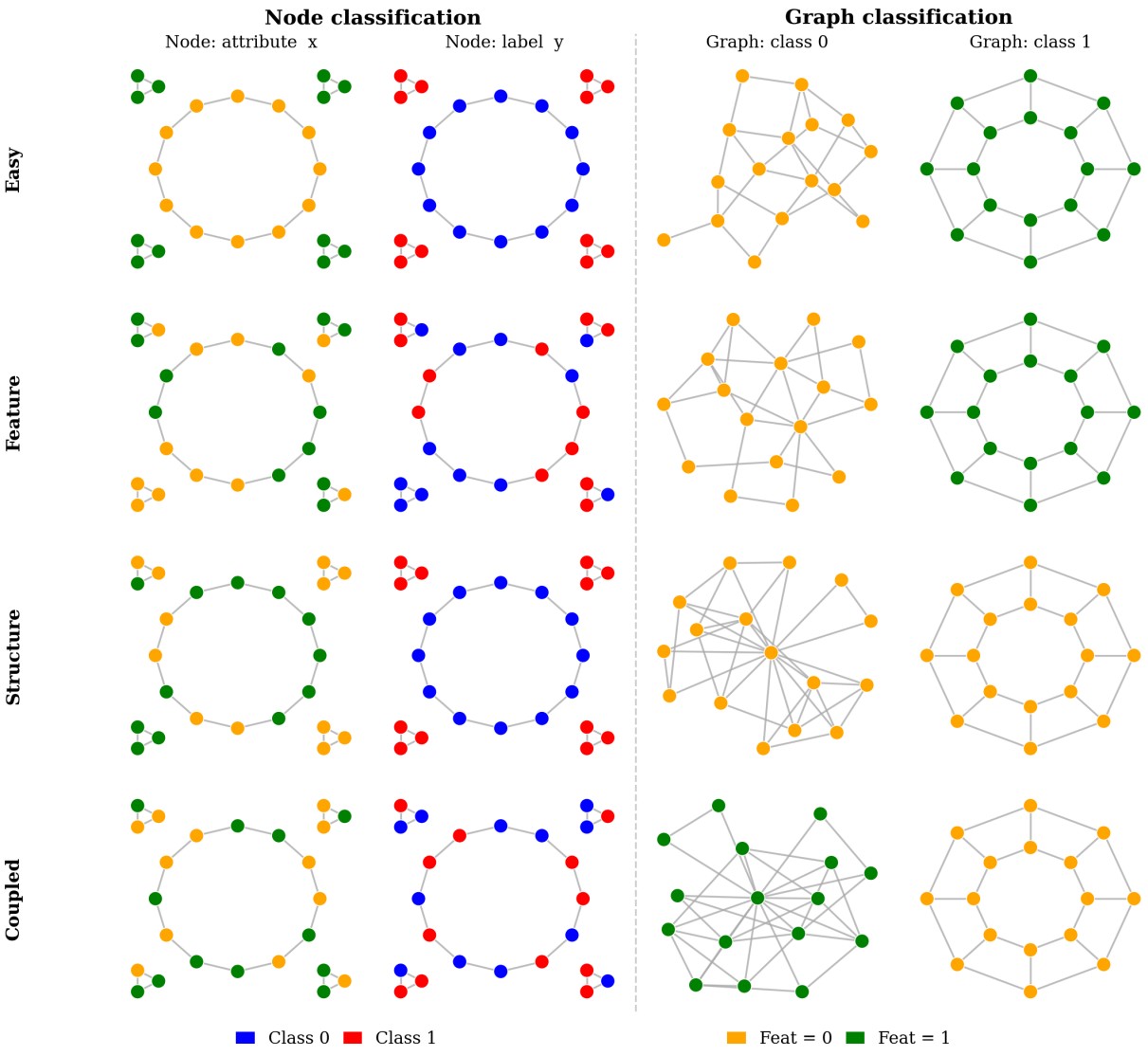

Figure 10: Examples of labelling schemes for our node-level and graph-level synthetic datasets.

