# OpenReview forum: "A Method and a Metric for GNN Reliance on Information from Features and Structure"
_TMLR — Rejected by TMLR_

### Review · Reviewer_8sVD · 2026-02-25

**Summary Of Contributions:**

The paper utilises noise-noise analysis to asses how much a GNN relies on features compared to graph structure.
The main idea is to test how much can each type of GNN be ignorant to one type of information and still retain similar performance.

To do this, they define the noise-noise ratio difference which compares how does the performance of the model degrade if a noise process corrupts the features versus how much it degrades if it corrupts the graph.
The noising process modifies an increasing proportion of graphs in the dataset by either replacing their node and edge features with random noise (sampled within the original feature range) while preserving structure, or replacing their graph structure with Erdős–Rényi graphs of the same edge count while preserving features, ensuring controlled and comparable degradation of information.


The results indicate that most common GNN layers are biased towards the features rather than the structure. Across both synthetic and real-world graph classification datasets, performance typically degrades more sharply under feature noise than under structural noise, suggesting a stronger reliance on feature information. Moreover, while all examined architectures can effectively ignore structure when it is uninformative, only GIN demonstrates consistent capacity to ignore features.

**Audience:**

Yes

**Audience Explanation:**

Overall, the paper is likely to be of interest to the TMLR audience, as it systematically examines the interplay between graph structure and node features in commonly used GNN architectures. Beyond contributing to methodological understanding, it can be helpful for practicioners ,e.g. , when features are not present and they consider the model they have to use and the feature construction procedure.

**Broader Impact Concerns:**

No evident impact concern.

**Claims And Evidence:**

Yes

**Claims Explanation:**

The claims are generally supported by systematic experiments on both synthetic datasets with controlled information sources and multiple real-world benchmarks, along with consistency and sensitivity analyses.

**Requested Changes:**

Overall the idea is meaningful but the paper should be more inclussive to be considered for publication.

- The current evaluation is limited to graph-level classification. Extending the analysis to node-level and edge-level prediction tasks would significantly strengthen the generality and impact of the method.

- The experiments are conducted on relatively small-to-moderate benchmarks. Additional evaluation on larger-scale graphs and higher-dimensional feature settings would help demonstrate the practicality and robustness of the approach.

- The method relies on specific assumptions about the feature and structure noising processes (e.g., replacing with Erdős–Rényi graphs and random features). It would be beneficial to assess how sensitive the conclusions are to alternative realistic perturbation schemes that are milder, such as edge rewiring and removing motifs or communities for the structure and Gaussian noise injection, permutation and dropout for the features.

---

> ### Author Response · Authors · 2026-03-06
> **Response to reviewer (1/3) - node classificaton**
>
> RESPONSE: Thank you for your measured and constructive review. We are glad that you found that our experiments supported our findings, systematic, and potentially useful for practitioners. Below we respond to each point of feedback in order, including suggested changes for the next revision, which we will prepare following the receipt of further reviews.
>
> ---
>
> ## R1.1 – Node-Level and Edge-Level Tasks
>
> **Reviewer concern:** The current evaluation is limited to graph-level classification. Extending the analysis to node-level and edge-level prediction tasks would significantly strengthen the generality and impact of the method.
>
> RESPONSE: Our method is indeed limited to graph-level tasks, and we appreciate that extension to node and edge-level tasks (i.e. node classification, edge prediction) might strengthen our contributions. Our method relies on paired noise on features and structure; for a node or edge-level task to remain meaningful and consistent under such paired noise, applied noise functions must be within-sample. Further, it is very difficult to design synthetic datasets (which are necessary to validate findings) under node and edge-level settings. Node classification is essentially predicting an additional node feature, and edge prediction similar for structural features, so designing tasks where these information sources do not conflate with labels is highly challenging.
>
> RESPONSE: As discussed in Section 3.1, within-sample noise functions are not well-justified for our use case. That said, graph-level tasks rely on the same methods as node and edge-level tasks, with the only difference being the final pooling operation. This means that our findings on graph-level tasks can appropriately be applied to node and edge-level ones.
>
> RESPONSE: There is a special case where node classification (or regression) does not rely on labels on other nodes, i.e. predictions are over all nodes in a graph. While we still risk the previously discussed conflation between information sources and labels, in this situation we are better justified in applying our whole-graph noise functions. We have, following this review, performed experiments along this line, making use of novel synthetic datasets in the same manner as our graph-level tasks. Worth noting is that due to the risks of conflation discussed above, these findings should be considered for completeness, with our graph-level findings remaining the principle contributions of this work.

---

> > ### Author Response · Authors · 2026-03-06
> > **Response to reviewer (2/3) - changes for node classification**
> >
> > CHANGES: Add the following to **Section 3.1**:
> >
> > > …might remove a ring structure. Crucially, the unsuitability of within-sample noise also precludes our approach from application to most node-level or edge-level tasks, though given the same mechanism for a given MPNN layer in both settings, our findings should apply here too.
> >
> > CHANGES: Add the following to **Section 5.1**:
> >
> > > **5.1.1 Node and Edge-Level Tasks**
> > > The necessity of using whole-graph noise functions precludes application in most node and edge-level tasks. In the limited scope that all node labels are predicted at once, there is no need for induction requiring the labels on other nodes, allowing us to perform experiments on node classification. This requires the development of new synthetic datasets with isolated information channels on a node level.
> > >
> > > We find that the results presented above are broadly valid in this level of granularity, although on the Easy dataset the use of information from features and structure is broadly balanced from all models. The GAT demonstrates heavy overfitting on features on the Structure dataset. An important note is that designing these synthetic datasets with full validity is more challenging on the node-level than the graph-level. Node classification is essentially predicting an additional node feature, and as a result, it is highly difficult to design synthetic datasets where information sources do not conflate with labels. Full experimental detail and results are presented in Appendix F.
> >
> > CHANGES: Add the following **Appendix F**:
> >
> > > The use of whole-sample noise functions makes extension to node or edge-level tasks highly challenging. This is particularly true of edge prediction, where it is not clear which edges *should* be present when all edges are random. Most node-level tasks are also excluded, as these commonly rely on labels present on other nodes in the graph, using principles like homophily.
> > >
> > > Within the limited case of predicting all node labels within a graph, we are able to extend our work to node classification, though developing synthetic datasets for this case is challenging. This is due to the difficulty of designing structure and feature patterns that do not overlap. As an example, for most possible structural features, simply summing incoming edges trivially solves the task. As a result we propose the following node-classification datasets with isolated information sources, visualised in Figure ??
> > >
> > > - **Easy** — Both information sources are predictive. Each node is assigned a binary label $y_i \in \{0,1\}$ uniformly at random. The node feature $x_i \in \{0.25, 0.5\}$ encodes $y_i$ directly, and class-1 nodes are wired into triangles (3-cliques) while class-0 nodes form a single cycle, so triangle membership also encodes $y_i$. All nodes have degree 2, ensuring that degree alone carries no signal.
> > >
> > > - **Feature-only** — Only node features are predictive. Labels and features are assigned as in **Easy**, but triangle membership is randomised independently of $y_i$, so local graph structure is uninformative.
> > >
> > > - **Structure-only** — Only graph structure is predictive. Triangle membership encodes $y_i$ as in **Easy**, but node features are drawn independently of the label, rendering them uninformative.
> > >
> > > - **Coupled** — Neither source is individually sufficient. Two independent balanced binary variables $a_i, b_i \sim \mathrm{Bern}(0.5)$ are drawn per node. The label is $y_i = \mathbf{1}[a_i = b_i]$; the feature encodes $a_i$ and triangle membership encodes $b_i$. Predicting $y_i$ therefore requires integrating both sources: neither feature nor structure alone is more than chance-level informative.
> > >
> > > These experiments are presented as a first step towards full extensions to node and edge-level tasks, and findings should be taken primarily from our graph-level results. Quantitative results are presented in Table ?? and visualised in Figure ??. Results broadly mirror those of our graph-level tasks, though the three layers evenly balance the use of feature and structure information on the Easy dataset. The GAT model heavily overfits on feature information on the Structure dataset, and no model achieves perfect performance even under zero noise on the Coupled dataset.

---

> > > ### Author Response · Authors · 2026-03-06
> > > **Response to reviewer (3/3) - Consistency w/size and dimensions, clarity on noise functions**
> > >
> > > ## R1.2 – Scale in Experiments
> > >
> > > **Reviewer concern:** The experiments are conducted on relatively small-to-moderate benchmarks. Additional evaluation on larger-scale graphs and higher-dimensional feature settings would help demonstrate the practicality and robustness of the approach.
> > >
> > > RESPONSE: We validate the consistency and invariance of Noise-Noise Analysis and NNRD w.r.t. graph size and feature dims in Appendix E, where we show that varying either has no significant impact on NNRD values. Here we evaluate $|V| \in \{0 \rightarrow 50, 50 \rightarrow 100, 100 \rightarrow 200, 200 \rightarrow 400\}$ and $X \in \mathbf{R}^d, d \in \{2, 5, 10, 25, 50\}$. We are happy to produce a wider range if necessary, and will add the following text to signpost this analysis more clearly.
> > >
> > > CHANGES: Add the following to the relevant section:
> > >
> > > > …Specifically, we make use of our own synthetic datasets as cases where information sources are well-understood, vary properties not relevant to this information, then verify statistically that NNRD is stable. The specific varied properties are node count, feature dimensionality and density where appropriate.
> > >
> > > ---
> > >
> > > ## R1.3 – Varied Noise Functions
> > >
> > > **Reviewer concern:** The method relies on specific assumptions about the feature and structure noising processes (e.g., replacing with Erdős–Rényi graphs and random features). It would be beneficial to assess how sensitive the conclusions are to alternative realistic perturbation schemes that are milder, such as edge rewiring and removing motifs or communities for the structure, and Gaussian noise injection, permutation and dropout for the features.
> > >
> > > RESPONSE: As discussed above, and in some detail in Section 3.1, we employ these noise functions precisely because they are harsh. Our aim is to remove all information from within a sample. Information within samples is rarely homogenous along a channel; one edge may carry significantly more pertinent information than another, and the same can be said for sensitivity to noise along individual numeric or categorical features. As a result, while these noise functions are invaluable in many other applications, and their extension for this method is a worthy area for future work, we do not believe them appropriate in this case. We will include an image to demonstrate the volatile destruction of information through functions like edge-level rewiring in the next revision.

---

### Review · Reviewer_fBXM · 2026-04-02

**Summary Of Contributions:**

This paper attempts to reveal the reliance of GNNs on different types of information. Specifically, the authors propose a metric called NNRD to explore the influence of different information on model performance. Experimental results on three classic GNNs seem to show promising results on revealing the reliance of attribute an topology information.

**Audience:**

Yes

**Audience Explanation:**

The paper focuses on an interesting topic of graph representation learning via GNN models.

**Claims And Evidence:**

Yes

**Claims Explanation:**

The paper provides a thorough theoretical analysis, along with detailed experimental validation across multiple datasets.

**Requested Changes:**

1.This paper provides a quantitative analysis on three typical GNNs. Though applying NNRD seems to be an interesting idea for GNN’s interpretability, I think the discussions about the experimental results are relative shallow. These three GNN (GCN, GAT and GIN) are built on different architecture designs. GCN utilizes the topology-oriented aggregation strategy, while GAT is based on feature-driven aggregation. Moreover, GIN adopts a flexible mechanism to fuse the information of node itself and its neighbors. The authors should provide deep insights on different architectures according to the obtained results.

2.How do you set “only structure points to the graph label” in Structure (Single-Source)? The detailed information about the experimental setting is required.

3.Actually, the pre-train stage could also affect the experimental results, such as the GNN layers, etc. Different architectures may prefer different parameters. Do you consider this situation?

4.The authors leverage the graph classification task to analyze the ability of GNNs on information integration which may be not very suitable. The core function of GNNs is still the representation learning on node-level. When the scenario transitions to node-level representation learning, the situation becomes more complex and meaningful. Hence, conducting analysis on node-level representation learning could be more valuable.

---

> ### Author Response · Authors · 2026-04-10
> **Response to Reviewer fBXM**
>
> ## fBXM.1 — Depth of Discussion on Architecture Differences
>
> We expanded Section 6 to make the architectural sources of the observed behavioural differences explicit, drawing on the details in Appendix A. In brief: the GCN entangles features and structure via a single shared weight matrix, with no mechanism to suppress either source. The GAT's attention scores are feature-driven, so when features are uninformative (Structure and Coupled datasets) the attention mechanism lacks a useful signal, explaining the GAT's failure on those tasks. The GIN's post-aggregation MLP separates structural summation from learned transformation, enabling feature suppression when structure is more informative.
>
> > **Changes:** We expanded Section 6 with: *"The architectural source of these behavioural differences is informative. The GCN's single shared weight matrix entangles features and structure during message-passing, providing no clean mechanism to suppress either source. For the GAT, these results are consistent with architectural design, given that the attention mechanism is feature-driven. When features are uninformative, the attention scores have no useful signal to condition on. The GIN's post-aggregation MLP cleanly separates structural summation from learned transformation, enabling it to suppress feature information when structure is more informative."*
>
> ---
>
> ## fBXM.2 — Construction of the Structure (Single-Source) Dataset
>
> The Structure dataset uses ring ladder graphs (label 1) and Erdős–Rényi graphs (label 0) of matching density. Node and edge features are assigned $+1$ or $-1$ uniformly at random, independently of the label, ensuring $I(y; X) = 0$ by construction. We expanded the dataset description in Section 4.1 accordingly; Figure 3 provides a visual illustration. This description also addresses part (b) of zTSK.3 from Reviewer zTSK.
>
> > **Changes:** We expanded the Structure dataset description in Section 4.1 to: *"**Structure (Single-Source)** Graphs are ring ladders or ER graphs as above. The label is determined solely by the structural class (ring ladder → 1, ER → 0). Node and edge features are assigned $+1$ or $-1$ uniformly at random, independently of the label, ensuring $I(y; X) = 0$. Only structural information is predictive of the label."*
>
> ---
>
> ## fBXM.3 — Effect of Pre-training and Hyperparameter Choices
>
> Our hyperparameters (3-layer GNNs, 100 hidden units, learning rate 0.001, batch size 256) match Bechler-Speicher et al. (2024), providing a principled anchor for direct comparison. The invariance analysis in Appendix E shows NNRD is stable across feature dimensionality ($d \in \{2, 5, 10, 25, 50\}$), graph size ($|V| \in \{20\text{–}50, 50\text{–}100, 100\text{–}200, 200\text{–}400\}$), and density, confirmed via Kruskal–Wallis and Levene testing (Table 5). We do not explicitly vary the number of GNN layers; however, the stability across these other properties provides strong evidence that NNRD is not sensitive to non-task-relevant architectural choices of this kind. We added a signposting sentence to Section 4.
>
> > **Changes:** We added a sentence to Section 4: *"Hyperparameters are chosen to match Bechler-Speicher et al. (2024) to allow direct comparison. Appendix E demonstrates that NNRD is stable across variations in feature dimensionality, graph size and density, providing evidence of robustness to non-task-relevant hyperparameter choices."*
>
> ---
>
> ## fBXM.4 — Node-Level Representation Learning
>
> We extended our work to node-level classification in the scope where whole-graph noise functions remain applicable: the case where all node labels within a graph are predicted simultaneously. We developed four synthetic node-level datasets (Easy, Feature, Structure, Coupled) analogous to our graph-level ones, with controlled isolation of feature and structural information at node level (binary node features encoding one source; triangle membership encoding the other).
>
> Results broadly mirror our graph-level findings: all models tend to rely on feature information, though on the Easy dataset the three architectures show more balanced use of both sources than at graph level. The GAT shows heavy overfitting on features on the Structure dataset. As noted in response to 8sVD.1, these results are presented as complementary; our graph-level findings remain the principal contribution. Full details and results are in new Appendix F, with a summary in new Section 5.1.1.
>
> > **Changes:** As described in response to 8sVD.1, we added new Section 5.1.1 summarising node-level results, new Appendix F detailing the node-level synthetic datasets and full experimental results, and Figure 3 illustrating node-level and graph-level dataset labelling schemes.

---

> > ### Comment · Reviewer_fBXM · 2026-05-04
> >
> > Thanks for your response. The revision has addressed my concerns and I have no further questions.

---

### Review · Reviewer_zTSK · 2026-04-04

**Summary Of Contributions:**

*Summary:*

This manuscript proposes a methodology for quantifying the degree to which a neural model defined on graphs is learning representations based on features or structure.

Given a dataset $D$ consisting of graphs $G$ with features $X$, topology $E$ (edge connectivity), and graph-level label $y$, the authors propose perturbing the dataset with progressively increasing noise levels. For a given noise level $t$, a proportion of graphs are subjected either to feature noise (replacing features with random noise) or structure noise (replacing graph connectivity with edge-count-preserving Erd\H{o}s--R'enyi graphs). A metric (NNRD) is then defined as an average log-ratio of performance under feature vs structure perturbation across noise levels, intended to quantify relative reliance on $X$ vs $E$.

Various synthetic and real-world experiments are conducted to evaluate NNRD. The synthetic experiments are designed to isolate settings where labels depend on features only, structure only, or both, and are used to study whether common GNN architectures (GCN, GIN, GAT) exhibit bias toward one source of information. The real-world experiments aim to show that these models often rely more heavily on features unless structural information is explicitly injected (e.g., via positional encodings).

*Strengths:*
* This manuscript explores the interesting and practically relevant question of feature–structure relationships in GNNs.
* The synthetic experiments appear well designed to probe distinct regimes (feature-only, structure-only, and joint dependence), which is a reasonable way to evaluate the proposed metric.
* The method is simple and can be applied post hoc to trained models.

*Weaknesses:*
* A primary concern is whether the dataset-level perturbation strategy actually produces comparable degradation of information across features and structure. While Section 3.1 motivates graph-level replacement as an alternative to fine-grained perturbations, it does not convincingly establish that replacing features with random noise and replacing connectivity with edge-count-preserving Erdos-Renyi graphs induce equal information loss. Moreover, the theoretical arguments in Appendices B and C rely on strong assumptions (e.g., that random features are uninformative and ER graphs contain negligible GNN-accessible information) that are not clearly justified in the experimental settings.
* There appears to be an incorrect mutual information decomposition in Appendix B. The paper assumes $I(y;\tilde X) = (1-t)I(y;X) + t I(y;X^{\text{rand}})$ for the feature-noised mixture, but mutual information is not linear under mixtures unless additional conditions hold. Since the equal-information-loss argument relies on this step, the theoretical justification is incomplete.
*There are several imprecisions in the discussion portions of the manuscript, e.g.:
    * Section 2.2 claims that “a compromise should be struck between different information sources in graph learning,” but the cited works (e.g., oversquashing and information bottleneck) do not directly support this specific statement.
    * The implicit assumption that replacing $E$ with ER graphs leads to near-complete loss of structural information may not hold for datasets whose informative structure is close to random-graph statistics.
    *More generally, the notion of “information” shifts between mutual information and what GNNs can actually extract, without clearly distinguishing the two.

**Audience:**

Yes

**Audience Explanation:**

The manuscript tackles the important problem of quantifying the relative importance of features and topology in representation learning in GNNs. NNRD is also shown to be easy to use for practitioners as it does not require retraining of a given model.

**Claims And Evidence:**

No

**Claims Explanation:**

The manuscript claims that NNRD is an _interpretable_ metric for quantifying the contribution of structural and feature information to GNN representation learning. However, the ability of the given algorithm to properly characterize information loss between $E$ and $X$ in a commensurate manner is not established. Moreover, discussion of the implications of the algorithm and results is sometimes imprecise, hurting the ability of the manuscript to validate claims. For example, results from a small set of datasets is extrapolated to general GNN behavior without proper justification.

**Requested Changes:**

* [critical] The manuscript should better justify the graph level perturbation strategy. Section 3.1 argues that feature perturbation and edge-swapping lead to inconsistent information loss yet it is not explained the mechanism by which the use of graph-level noising functions can overcome this issue.
* [critical] Similar to above, the manuscript should justify how the replacement of edge connectivity information with Erdos Renyi (ER) graphs leads to uniform topological information loss in the general case? What about datasets or domains where topology is more ER-like than others? What is the significance of using ER graphs over other random graph generators? Analogously, it not clear how the replacement of features with "random noise" uniformly removes information in the general case? E.g. what about cases where the original node information is closer in distribution to the random noise generator used than others?
* [critical] The manuscript contains some underspecified discussions. The presentation would improve if specificity was added. For example:
    * In Section 3.1, feature replacement is described as replacing features with "random noise." Is this Guassian noise? Uniform noise following the support of the features in the training set? Something else?
    * In Section 4.1, the method by which the various datasets (easy, feature, structure, coupled) are created is not stated. Especially in the case of the "Coupled" dataset, this construction seems non-trivial, so additional details would improve understanding of the experimental results substantially.
* [critical] The mutual information decomposition in Appendix B should be revisited to justify if/when MI is linear wrt to mixtures.
* [minor] Depending on the particular distribution used to sample noise for the FeatureNoise function, the manuscript may benefit from discussion of cases where such (continuous) random noise may actually help performance of GNNs due to increase in expressiveness [1]. This may be a confounding variable in the ultimate performance analysis.
* [minor] Some claims made in the paper would benefit from relevant citations from the literature. For example, Section 4.2 claims that Enzyme and Protein datasets have been used as examples of domains with uninformative structures but a citation is not provided.
* [minor] While the notion that degrading either $X$ or $E$ necessarily degrades their interdependence is briefly acknowledged is Section 3, the manuscript could benefit from additional analysis/discussion. How does this affect the conclusions reached in Section 5 and 6?
* [minor] The manuscript should better motivate the notions of GNN expressiveness and oversquashing mentioned in Section 2.2 relate directly to the thesis that "a compromise must be struct between different information sources in graph learning." This seems like a non-trivial conclusion to the best of my knowledge.


[1] Sato, Ryoma, Makoto Yamada, and Hisashi Kashima. "Random features strengthen graph neural networks." Proceedings of the 2021 SIAM international conference on data mining (SDM). Society for Industrial and Applied Mathematics, 2021.

---

> ### Author Response · Authors · 2026-04-10
> **Response to Reviewer zTSK (1)**
>
> ## zTSK.1 — Justification of Graph-Level Perturbation Strategy *(critical)*
>
> The justification for graph-level replacement is more explicit in Appendix B than in Section 3.1; we added a forward reference to make this connection clear. The core argument is as follows. Within-sample perturbation methods (e.g. edge-swapping, feature dropout) fail to satisfy Equation 6 because each perturbation step destroys an uncontrollable and unverifiable amount of information — as illustrated in Figure 1, a single broken edge may remove a ring structure entirely. There is no tractable way to prove that such methods degrade information at equal rates across the two channels. Graph-level replacement avoids this problem by construction: at noise level $t$, each replaced graph contributes *either* its full original information *or* effectively zero information (Lemmas 1 and 2). The mixing is Bernoulli-indexed, which means the proportional loss rate is exactly $-1$ for both channels (Theorem 3), directly satisfying Equation 6. This all-or-nothing design is what makes the performance curves directly comparable.
>
> > **Changes:** We added a forward reference to Section 3.1: *"The theoretical justification for why graph-level replacement achieves equal proportional information loss, while within-sample methods do not, is provided in Appendix B (Theorem 3)."*
>
> ---
>
> ## zTSK.2 — Uniformity of Information Loss under ER Graphs and Random Features *(critical)*
>
> Even if a dataset's structure is already near-ER, any structural signals in the original graphs are replaced by a freshly sampled ER graph, destroying any correlation with the label. NNRD therefore correctly reflects that structural information is low in such datasets — an accurate finding, not a false negative. The same argument holds for features close in distribution to the noise generator: Lemma 1 bounds spurious correlations at $O(d \log(N)/N)$, which is negligible for typical dataset sizes, and Appendix E confirms stability across feature dimensionalities. We added a clarifying note to Section 3.1 covering both points.
>
> > **Changes:** We added a clarifying note to Section 3.1 after the description of noise functions: *"In datasets where original structure is close to ER graphs, or original features are close in distribution to the noise generator, the noise functions still replace the original information entirely — any correlation with labels is lost upon replacement. NNRD will correctly reflect low reliance on the noised channel in such cases, which is an accurate characterisation of the dataset rather than a methodological failure."*
>
> ---
>
> ## zTSK.3 — Underspecified Discussions *(critical)*
>
> (a) Feature noise samples uniformly within the per-feature range of the training set. We clarified this in Section 3.1.
>
> (b) All synthetic datasets use graphs that are either ring ladders or Erdős–Rényi graphs, with node and edge features of $+1$ or $-1$. The Coupled dataset assigns a positive label when the structural class and feature value agree, and negative otherwise, making neither source alone sufficient but their combination fully predictive. We expanded the description in Section 4.1; a visual illustration is provided in Figure 3.
>
> > **Changes:** In Section 3.1, we replaced "random noise" with "noise sampled uniformly within the per-feature range of the training set."
>
> > **Changes:** In Section 4.1, we expanded the Coupled dataset description to: *"**Coupled (Both Sources)** Graphs are ring ladders or ER graphs as above, with features of $+1$ or $-1$. The label is positive when the structural class and the feature value agree (i.e. ring ladder with $+1$, or ER with $-1$), and negative otherwise. Neither source alone carries information about the label, but their combination is fully predictive, requiring the model to integrate both channels."*
>
> ---

---

> ### Author Response · Authors · 2026-04-10
> **Response to Reviewer zTSK (2)**
>
> ## zTSK.4 — Mutual Information Decomposition in Appendix B *(critical)*
>
> Mutual information is not in general linear under mixtures, but our construction is a specific Bernoulli-indexed mixture. At noise level $t$, each graph independently retains its original information (with probability $1-t$) or is replaced by independently sampled noise (with probability $t$). Letting $Z_i \sim \text{Bern}(t)$ indicate whether graph $i$ is noised, the law of total expectation gives:
>
> $$I(y;\, \mathcal{N}^X_t(X)) = \mathbb{E}_{Z}[I(y; X \mid Z)] = (1-t)\cdot I(y; X) + t \cdot I(y; X^{\text{rand}})$$
>
> Since $X^{\text{rand}}$ is generated independently of $y$, $I(y; X^{\text{rand}}) = 0$ by Lemma 1, recovering Equation 15. The key condition is the independence of the noise process from the labels, which holds by construction. We revised Appendix B.1 to make this argument explicit.
>
> > **Changes:** We revised the opening of Appendix B.1 to include: *"The decomposition in Equation 15 holds because the mixing is Bernoulli-indexed and independent of labels. Let $Z_i \sim \text{Bern}(t)$ indicate whether graph $i$ is noised. By the law of total expectation over $Z$:*
> > $$I(y;\, \mathcal{N}^X_t(X)) = (1-t)\cdot I(y;X) + t\cdot I(y;X^{\text{rand}})$$
> > *Since $X^{\text{rand}}$ is independent of $y$ by construction, $I(y;X^{\text{rand}}) = 0$ (Lemma 1), and the decomposition follows."*
>
> ---
>
> ## zTSK.5 — Random Features and GNN Expressiveness *(minor)*
>
> Random node features can increase GNN expressiveness by breaking symmetry (Sato 2021). However, in our setup noise is applied only at test time to a pre-trained model. Since the model's weights are fixed and were not trained to exploit random features, any expressiveness gains are irrelevant. Any spurious correlation between random features and labels is bounded by Lemma 1. We added a note to Section 3.1 to this effect.
>
> > **Changes:** We added a note to Section 3.1: *"We note that random node features have been shown to increase GNN expressiveness [Sato 2021]. However, as noise is applied only at test time to a pre-trained model, any such expressiveness gains cannot be exploited — the model's weights are fixed and were not trained to make use of random features."*
>
> ---
>
> ## zTSK.6 — Missing Citations *(minor)*
>
> We added citations to Errica 2022 and Bechler-Speicher 2024 at the relevant point in Section 4.2, both of which use these datasets in exactly this context.
>
> > **Changes:** We added citations to Errica 2022 and Bechler-Speicher 2024 in Section 4.2 where Enzymes and Proteins are described as examples of datasets with uninformative structure.
>
> ---
>
> ## zTSK.7 — Interdependence of Channels Under Noise *(minor)*
>
> Degrading either channel also degrades their joint mutual information $I(y; X, E)$, meaning NNRD reflects a model's relative reliance on each channel rather than their fully isolated contributions. We acknowledge this as a limitation. The Coupled dataset is designed to probe this case directly, and results there show NNRD behaves sensibly under joint dependence, with neither channel dominating. We added a caveat to Section 3 and a note on future directions to Section 7.
>
> > **Changes:** We added to Section 3: *"A caveat is that degrading either channel also degrades their joint mutual information $I(y; X, E)$. NNRD therefore reflects a model's relative reliance on each channel, rather than their fully isolated contributions. The Coupled dataset probes this case directly, requiring integration of both sources for accurate prediction."*
>
> > **Changes:** We added to Section 7: *"A fuller treatment of how channel interdependence affects NNRD is an open theoretical question and a natural direction for future work."*
>
> ---
>
> ## zTSK.8 — Motivation for Expressiveness and Oversquashing Claims *(minor)*
>
> The reviewer is correct that the cited works on expressiveness and oversquashing do not directly support the claim that a compromise must be struck between feature and structural information sources. We tightened this claim in Section 2.2, moving Wang 2024 and Davies 2024 forward to support it directly, and softened the reference to oversquashing and expressiveness to reflect what those works actually show.
>
> > **Changes:** In Section 2.2, we revised the relevant claim to: *"Some works explicitly decouple features and structure, finding performance benefits despite the loss of information from their inter-relation [Wang 2024, Davies 2024], suggesting that for some tasks one source may actively inhibit learning from the other."* We softened the reference to oversquashing and expressiveness, which do not directly support this specific claim.

---

### Author Response · Authors · 2026-04-10
**Response to All Reviewers**

We thank all three reviewers for their careful and constructive engagement with our work. We are encouraged by the broadly positive reception of the core idea and experimental design, and have made substantive revisions in response to the concerns raised. This section summarises the strengths and weaknesses identified across reviews, paired with the changes we made. Detailed responses to individual points follow in the per-reviewer sections below.

---

## Strengths Identified

Across all three reviews, the following aspects were received positively:

- The problem is practically relevant and interesting: all three reviewers agreed the paper addresses a meaningful question about feature–structure balance in GNNs (8sVD, zTSK, fBXM).
- The synthetic experimental design was praised for its principled construction and controlled isolation of information sources (8sVD, zTSK, fBXM).
- The method's post-hoc applicability to pre-trained models without retraining was highlighted as a practical strength (zTSK, fBXM).
- The systematic evaluation across both synthetic and real-world datasets, along with the consistency and sensitivity analyses, was noted as supporting the claims (8sVD).

---

## Weaknesses and Responses

**Scope limited to graph-level tasks (8sVD.1, fBXM.4).** Both Reviewers 8sVD and fBXM requested extension to node and edge-level tasks. We have extended our work to node-level classification in the limited scope where whole-graph noise functions remain applicable — the case where all node labels within a graph are predicted simultaneously. We developed four synthetic node-level datasets (Easy, Feature, Structure, Coupled) directly analogous to our graph-level ones. Results broadly mirror our graph-level findings: all models exhibit a feature bias, the GAT heavily overfits on features on the Structure dataset, and no model achieves perfect performance on the Coupled dataset. We note that full node and edge-level extension remains non-trivial (within-sample noise is not well-justified, and isolating information sources at node level is harder by construction); these results are therefore presented as complementary to the primary graph-level findings. Full details are in new Appendix F, with a summary in new Section 5.1.1.

**Theoretical justification of noise functions (zTSK.1, zTSK.2, zTSK.4).** Reviewer zTSK raised three critical concerns about the theoretical foundations: the justification for graph-level perturbation over within-sample methods; the uniformity of information loss under ER replacement and random features; and the validity of the mutual information decomposition in Appendix B. We addressed all three. We added a forward reference in Section 3.1 to Appendix B (Theorem 3), which proves that graph-level replacement achieves equal proportional information loss by construction. We added a clarifying note on the self-correcting behaviour of ER replacement when datasets are already near-ER. We revised Appendix B.1 to make the Bernoulli-indexed mixture argument explicit, showing that the decomposition follows from the law of total expectation, with label-independence of the noise process as the key enabling condition.

**Underspecified experimental and methodological details (zTSK.3, fBXM.2).** We clarified in Section 3.1 that feature noise is sampled uniformly within the per-feature range of the training set. We expanded the synthetic dataset descriptions in Section 4.1, including a detailed construction of the Coupled and Structure datasets. Figure 3 provides a visual illustration of labelling schemes across all datasets.

**Shallow architectural discussion (fBXM.1).** We expanded Section 6 to explicitly connect the observed behavioural differences to the architectural properties of GCN, GAT and GIN, drawing on the details in Appendix A.

**Imprecisions in related work claims (zTSK.8).** We tightened the claim in Section 2.2 and moved Wang 2024 and Davies 2024 forward as direct support, softening the reference to oversquashing and expressiveness.

**Scale and hyperparameter sensitivity (8sVD.2, fBXM.3).** Appendix E demonstrates stability of NNRD across feature dimensionality, graph size, and density. Hyperparameters match Bechler-Speicher et al. (2024) to enable direct comparison; we added a signposting sentence to Section 4.

---

### Decision · Action_Editor_Mp2U · 2026-06-03

**Recommendation:** Reject

**Additional Comments:**

This submission was reviewed by three expert reviewers. The reviewers agreed that the paper's topic is important and that it deserves further investigation. They also mentioned that the proposed approach could prove useful for practitioners. However, they also raised several concerns in the reviews, mainly about the following: (i) the method is limited to graph-level tasks; (ii) the employed noise functions are overly extreme since they remove all information from a sample; (iii) the real-world datasets used in the evaluation are relatively small; (iv) the empirical results are discussed only briefly; and (v) important details regarding the type and implementation of the injected noise are missing. More importantly, Reviewer zTSK raised significant concerns regarding the soundness of the proposed analysis, as discussed above. While the authors addressed some of the reviewers' comments in the revision, the concerns related to soundness remain unresolved.

Following the authors' response, two reviewers lean toward rejection, while one reviewer leans toward acceptance. Based on the remaining concerns regarding soundness, I recommend rejection of the submission at this time. However, I believe that the overall research direction is interesting and potentially valuable. I therefore suggest the authors address the identified issues and submit a major revision in the future.

**Audience:**

Yes

**Audience Explanation:**

In my view, the paper is likely to be of interest to the TMLR community since it studies the reliance of GNN models on information from the node features and the graph structure, an important problem in graph machine learning.

**Claims And Evidence:**

No

**Claims Explanation:**

The submission makes the following key claims:

* It introduces the so-called Noise-Noise Analysis which is intended to quantify the relative contributions of node features and graph structure to the performance of GNN models. However, some of the theoretical results accompanying the proposed analysis appear to be incorrect. In particular, Reviewer zTSK argues that the equality in Equation (15) does not generally hold, since mutual information is not linear under mixtures. I share this concern and I think that the equality might only hold as an inequality.

* It proposes the Noise-Noise Ratio Difference (NNRD), a metric whose objective is to quantify whether a GNN relies more heavily on node features or graph structure. Reviewer zTSK questioned the soundness of this metric, since it is not clear whether the noise injected into node features is directly comparable to the noise introduced into the graph structure. Specifically, it is unclear whether replacing node features with random noise and replacing the graph structure with Erdős–Rényi graphs containing the same number of edges both result in equivalent levels of information loss.

* The paper claims that GCN, GAT, and GIN layers are all capable of graph-less learning, while only GIN can perform feature-less learning. It also claims that all three architectures exhibit a bias toward node features over graph structure. These claims are supported by the experimental results presented in the paper.

**Resubmission Of Major Revision:**

The authors may consider submitting a major revision at a later time.